# Directed evolution of the rRNA methylating enzyme Cfr reveals molecular basis of antibiotic resistance

Kaitlyn Tsai[1], Vanja Stojković[1], Lianet Noda-Garcia[2], Iris D Young[3], Alexander G Myasnikov[4], Jordan Kleinman[1], Ali Palla[1], Stephen N Floor[5,6], Adam Frost[4,7], James S Fraser[3,7], Dan S Tawfik[2], Danica Galonić Fujimori[1,7,8]*

[1]Department of Cellular and Molecular Pharmacology, University of California San Francisco, San Francisco, United States; [2]Department of Biomolecular Sciences, Weizmann Institute of Science, Rehovot, Israel; [3]Department of Bioengineering and Therapeutic Sciences, University of California San Francisco, San Francisco, United States; [4]Department of Biochemistry and Biophysics, University of California San Francisco, San Francisco, United States; [5]Helen Diller Family Comprehensive Cancer Center, University of California San Francisco, San Francisco, United States; [6]Department of Cell and Tissue Biology, University of California San Francisco, San Francisco, United States; [7]Quantitative Biosciences Institute, University of California San Francisco, San Francisco, United States; [8]Department of Pharmaceutical Chemistry, University of California San Francisco, San Francisco, United States

**Abstract** Alteration of antibiotic binding sites through modification of ribosomal RNA (rRNA) is a common form of resistance to ribosome-targeting antibiotics. The rRNA-modifying enzyme Cfr methylates an adenosine nucleotide within the peptidyl transferase center, resulting in the C-8 methylation of A2503 ($m^8A2503$). Acquisition of *cfr* results in resistance to eight classes of ribosome-targeting antibiotics. Despite the prevalence of this resistance mechanism, it is poorly understood whether and how bacteria modulate Cfr methylation to adapt to antibiotic pressure. Moreover, direct evidence for how $m^8A2503$ alters antibiotic binding sites within the ribosome is lacking. In this study, we performed directed evolution of Cfr under antibiotic selection to generate Cfr variants that confer increased resistance by enhancing methylation of A2503 in cells. Increased rRNA methylation is achieved by improved expression and stability of Cfr through transcriptional and post-transcriptional mechanisms, which may be exploited by pathogens under antibiotic stress as suggested by natural isolates. Using a variant that achieves near-stoichiometric methylation of rRNA, we determined a 2.2 Å cryo-electron microscopy structure of the Cfr-modified ribosome. Our structure reveals the molecular basis for broad resistance to antibiotics and will inform the design of new antibiotics that overcome resistance mediated by Cfr.

*For correspondence: danica.fujimori@ucsf.edu

## Editor's evaluation

The paper addresses an important unresolved mechanism of antibiotic resistance caused by a RNA-modifying enzyme Cfr, a protein that confers resistance to multiple ribosome-targeting antibiotics due to methylation of the rRNA residue A2503 on the large ribosomal subunit. Tsai et al. identify the mutational hotspots that increase Cfr activity and show how a single methyl group on A2503 can hamper the antibiotic binding. The paper is an important contribution to our understanding of antibiotic resistance and is of broad interest to readers in the field of antibiotic resistance, biochemistry, structural biology, and medicinal chemistry.

## Introduction

A large portion of clinically relevant antibiotics halt bacterial growth by binding to the ribosome and inhibiting protein synthesis (*Tenson and Mankin, 2006*; *Wilson, 2009*; *Arenz and Wilson, 2016*). Since antibiotic binding sites are primarily composed of ribosomal RNA (rRNA), rRNA-modifying enzymes that alter antibiotic binding pockets are central to evolved resistance (*Vester and Long, 2013*; *Wilson, 2014*). The rRNA-methylating enzyme Cfr modifies an adenosine nucleotide located within the peptidyl transferase center (PTC), a region of the ribosome essential for catalyzing peptide bond formation and consequently, a common target for antibiotics (*Schwarz et al., 2000*; *Kehrenberg et al., 2005*). Cfr is a radical SAM enzyme that methylates the C8 carbon of adenosine at position 2,503 (m$^8$A2503, *Escherichia coli* numbering) (*Jensen et al., 2009*; *Kaminska et al., 2010*; *Yan et al., 2010*; *Yan and Fujimori, 2011*; *Grove et al., 2011*). Due to the proximal location of A2503 to many antibiotic binding sites, introduction of a single methyl group is sufficient to cause resistance to eight classes of antibiotics simultaneously: **ph**enicols, **l**incosamides, **o**xazolidinones, **p**leuromutilins, **s**treptogramin **A** (PhLOPS$_A$), in addition to nucleoside analog A201A, hygromycin A, and 16-membered macrolides (*Long et al., 2006*; *Smith and Mankin, 2008*; *Polikanov et al., 2015*). Among rRNA modifying enzymes, this extensive cross-resistance phenotype is unique to Cfr and presents a major clinical problem.

Cfr emergence in human pathogens appears to be a recent event, with the first case reported in 2007 from a patient-derived *Staphylococcus aureus* isolate (*Toh et al., 2007*; *Arias et al., 2008*). Since then, the *cfr* gene has been identified across the globe in both gram-positive and gram-negative bacteria, including *E. coli* (*Shen et al., 2013*; *Vester, 2018*) and has been associated with several clinical resistance outbreaks to the oxazolidinone antibiotic, linezolid (*Morales et al., 2010*; *Locke et al., 2010*; *Bonilla et al., 2010*; *Cai et al., 2015*; *Layer et al., 2018*; *Lazaris et al., 2017*; *Dortet et al., 2018*; *Weßels et al., 2018*). The vast spread of Cfr is attributed to its association with mobile genetic elements and relatively low impact on bacterial fitness, suggesting that *cfr* can be rapidly disseminated within bacterial populations (*LaMarre et al., 2011*; *Schwarz et al., 2016*).

Due to the ability of Cfr to confer resistance to several antibiotics simultaneously, it is critical to understand how bacteria may adapt under antibiotic pressure to enhance Cfr activity and bolster protection against ribosome-targeting molecules. Identification of Cfr mutations that improve resistance will also be critical for informing clinical surveillance and designing strategies to counteract resistance. A major limitation in our current understanding of Cfr-mediated resistance is the lack of structural insight into changes in the ribosome as a result of Cfr modification. Steric occlusion of antibiotic binding has been proposed as a model to rationalize altered antibiotic susceptibility (*Polikanov et al., 2015*). Additionally, the observation that A2503 can adopt both *syn* and *anti*-conformations in previously reported ribosome structures suggests that methylation may regulate conformation of the base, as previously proposed (*Toh et al., 2008*; *Schlünzen et al., 2001*; *Tu et al., 2005*; *Stojković et al., 2020*). However, direct evidence for how m$^8$A2053 alters antibiotic binding sites to inform the design of next-generation molecules that can overcome Cfr resistance is lacking.

In this study, we identified mechanisms that enhance antibiotic resistance by performing directed evolution of a *cfr* found in a clinical MRSA isolate under antibiotic selection (*Barlow and Hall, 2003*). The obtained highly resistant Cfr variants show increased rRNA methylation, driven primarily by robust improvements in Cfr cellular levels, achieved either by higher transcription or increased translation and improved cellular stability. In particular, mutation of the second Cfr amino acid to lysine strongly enhances translation and resistance. Finally, we used an evolved variant which achieves near-stoichiometric rRNA methylation to generate a high-resolution cryo-electron microscopy (EM) structure of the Cfr-modified *E. coli* ribosome. The obtained structural insights provide a rationale for how m$^8$A2503 causes resistance to ribosome antibiotics.

## Results

### Evolved Cfr variants confer enhanced antibiotic resistance

To perform directed evolution of Cfr, we used error-prone PCR (EP-PCR) to randomly introduce 1–3 mutations into the *cfr* gene obtained from a clinical MRSA isolate (*Toh et al., 2007*), herein referred to as CfrWT (*Figure 1a*). Mutagenized *cfr* sequences were then cloned into a pZA vector where Cfr was expressed under tetracycline-inducible promoter P$_{tet}$ introduced to enable precise control of Cfr

**eLife digest** Antibiotics treat or prevent infections by killing bacteria or slowing down their growth. A large proportion of these drugs do this by disrupting an essential piece of cellular machinery called the ribosome which the bacteria need to make proteins. However, over the course of the treatment, some bacteria may gain genetic alterations that allow them to resist the effects of the antibiotic.

Antibiotic resistance is a major threat to global health, and understanding how it emerges and spreads is an important area of research. Recent studies have discovered populations of resistant bacteria carrying a gene for a protein named chloramphenicol-florfenicol resistance, or Cfr for short. Cfr inserts a small modification in to the ribosome that prevents antibiotics from inhibiting the production of proteins, making them ineffective against the infection. To date, Cfr has been found to cause resistance to eight different classes of antibiotics. Identifying which mutations enhance its activity and protect bacteria is vital for designing strategies that fight antibiotic resistance.

To investigate how the gene for Cfr could mutate and make bacteria more resistant, Tsai et al. performed a laboratory technique called directed evolution, a cyclic process which mimics natural selection. Genetic changes were randomly introduced in the gene for the Cfr protein and bacteria carrying these mutations were treated with tiamulin, an antibiotic rendered ineffective by the modification Cfr introduces into the ribosome. Bacteria that survived were then selected and had more mutations inserted. By repeating this process several times, Tsai et al. identified 'super' variants of the Cfr protein that lead to greater resistance.

The experiments showed that these variants boosted resistance by increasing the proportion of ribosomes that contained the protective modification. This process was facilitated by mutations that enabled higher levels of Cfr protein to accumulate in the cell. In addition, the current study allowed, for the first time, direct visualization of how the Cfr modification disrupts the effect antibiotics have on the ribosome.

These findings will make it easier for clinics to look out for bacteria that carry these 'super' resistant mutations. They could also help researchers design a new generation of antibiotics that can overcome resistance caused by the Cfr protein.

expression (*Wellner et al., 2013*). The resulting library of $\sim 10^7$ *E. coli* transformants was selected for growth in the presence of increasing amounts of tiamulin, a pleuromutilin antibiotic to which Cfr confers resistance. During each round, a subset of the surviving colonies was sequenced to identify new mutations. After two rounds of evolution, wild-type Cfr was no longer detected, indicating that the introduced mutations provide enhanced survivability in the presence of tiamulin. After five rounds of mutation and selection, we performed two rounds of selection without mutagenesis, and with high tiamulin concentrations, thus leading to fixation of mutations that provide robust resistance.

Analysis of surviving *cfr* sequences from the final rounds of selection revealed notable trends (*Supplementary file 1*). Three positions were primarily mutated: N2, I26, and S39. By homology modeling, these mutational hotspots appear distal from the enzyme active site (>12 Å; *Figure 1b*). In fact, these mutations reside in what has been predicted to be an N-terminal accessory domain separate from the radical-SAM catalytic domain (*Kaminska et al., 2010*). Second, ~28% of sequences contained alterations to the promoter. These alterations consist of either $P_{tet}$ duplication, or insertion of a partial $P_{tet}$ sequence (*Supplementary file 1*).

We selected seven evolved Cfr variants, referred herein as CfrV1–V7, as representative mutational combinations for further characterization (*Figure 1c*). All selected Cfr variants contain mutations in the *cfr* open reading frame (ORF) while CfrV6 and CfrV7 also harbor $P_{tet}$ alterations (*Figure 1d*). Compared to CfrWT, these variants confer ~2-fold to ~16-fold enhanced resistance to PhLOPS$_A$ antibiotics and hygromycin A, with no changes in susceptibility to trimethoprim, an antibiotic that does not inhibit the ribosome (*Figure 1e*, *Figure 1—figure supplement 1*). Interestingly, the promoter alterations enable CfrV7 to be expressed and confer resistance to tiamulin in the absence of inducer (*Figure 1—figure supplement 2*). The robustness of resistance, and the absence of active-site mutations, suggests Cfr variants do not act as dominant-negative enzymes that inhibit C-2 methylation of A2503, as observed in a previous directed evolution experiment (*Stojković et al., 2016*). Furthermore, the specificity of resistance to PhLOPS$_A$ antibiotics suggests that these Cfr variants

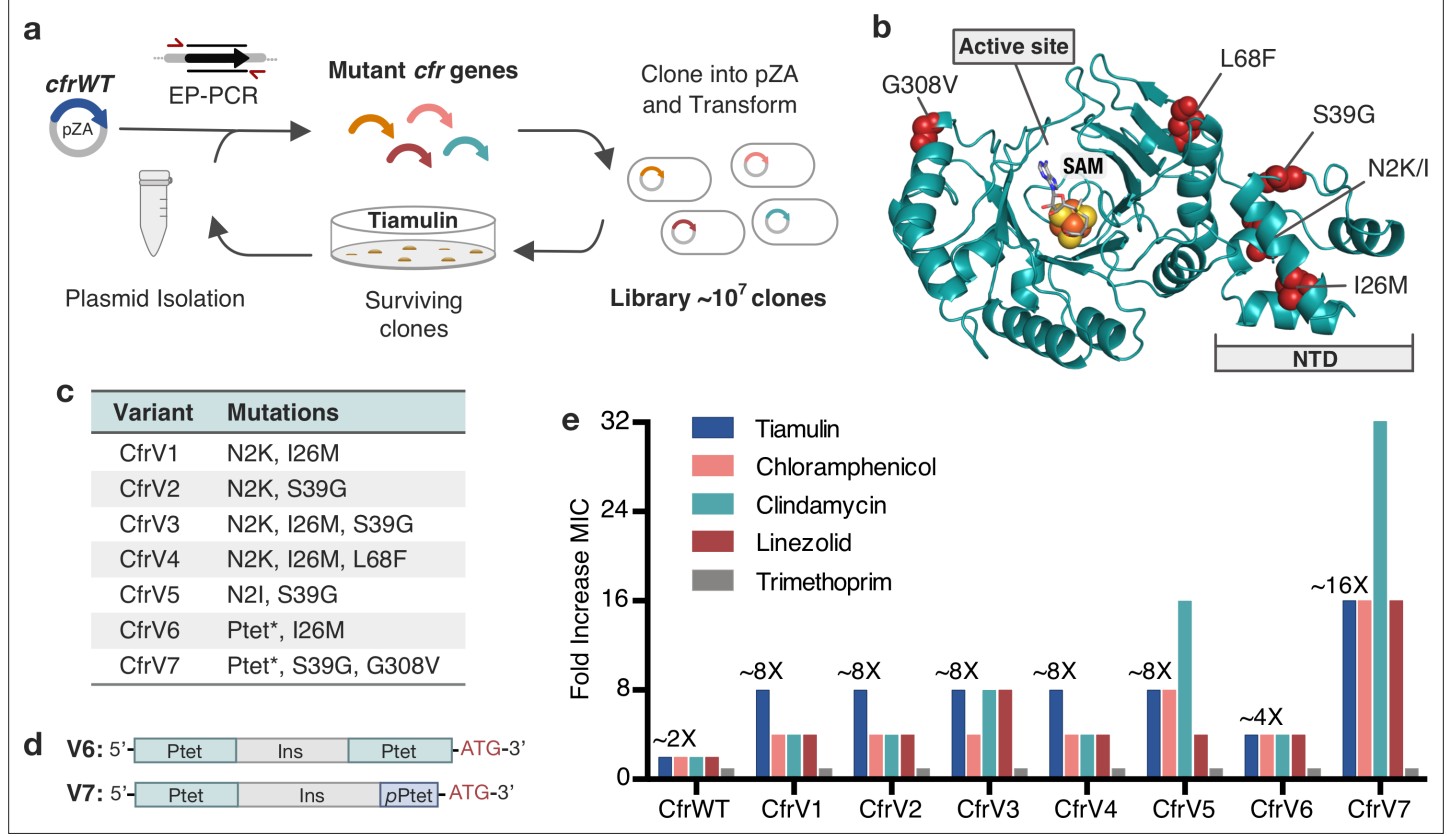

**Figure 1.** Evolved variants of Cfr exhibit improved resistance to PhLOPS$_A$ ribosome antibiotics. (**a**) Evolution of Cfr under selection by the PTC-targeting antibiotic tiamulin. (**b**) Cfr homology model based on RlmN generated by I-TASSER server (*Yang and Zhang, 2015*) with mutagenic hotspots in red. N-terminal domain (NTD) is labeled. Active site denoted by *S*-adenosylmethionine (SAM, gray) and [4Fe-4S] cluster (orange). (**c**) Evolved variants containing Cfr mutations were selected for further study. Ptet* indicates alterations to promoter sequence. (**d**) Promoter architecture of CfrV6 and CfrV7 where *p*Ptet designates a partial Ptet promoter sequence and Ins designates a variable insertion sequence. (**e**) Fold improvement in MIC resistance value for PhLOPS$_A$ antibiotics and trimethoprim compared to empty pZA vector control determined from three biological replicates by microbroth dilution method. Trimethoprim is a negative control antibiotic that does not target the ribosome. LZD testing was performed against *Escherichia coli* BW25113 lacking efflux pump, *acrB*. Numerical MIC values are displayed in *Figure 1—source data 1*.

The online version of this article includes the following figure supplement(s) for figure 1:

**Source data 1.** MIC numerical data.

**Source data 2.** Blot images.

**Figure supplement 1.** Cfr variants confer increased resistance to hygromycin A.

**Figure supplement 2.** CfrV7 does not require an inducer for resistance or expression.

elicit their effects through PTC modification rather than triggering a stress response that confers global resistance.

## Variants exhibit increased rRNA methylation and Cfr protein levels

To test the hypothesis that Cfr variants mediate higher resistance by increasing the fraction of ribosomes with m$^8$A2503, we evaluated the methylation status of A2503 by mass spectrometry. Specifically, we expressed Cfr in *E. coli* and used oligonucleotide protection to isolate a 40-nt fragment of 23S rRNA containing A2503 (*Andersen et al., 2004*; *Stojković and Fujimori, 2015*). The isolated fragment was then enzymatically digested and analyzed by MALDI-TOF mass spectrometry (*Figure 2a*, *Figure 2—figure supplement 1*). As expected, an empty vector produces a 1013 m/z fragment corresponding to the mono-methylated m$^2$A2503, modification installed by the endogenous enzyme RlmN. Upon expression of Cfr, we observe a reduction in the 1013 m/z peak and the emergence of a new peak at 1027 m/z, corresponding to m$^2$A2503 conversion into hypermethylated m$^2$m$^8$A2503. CfrWT is able to convert less than ~40% of m$^2$A2503 into the hypermethylated m$^2$m$^8$A2503 product.

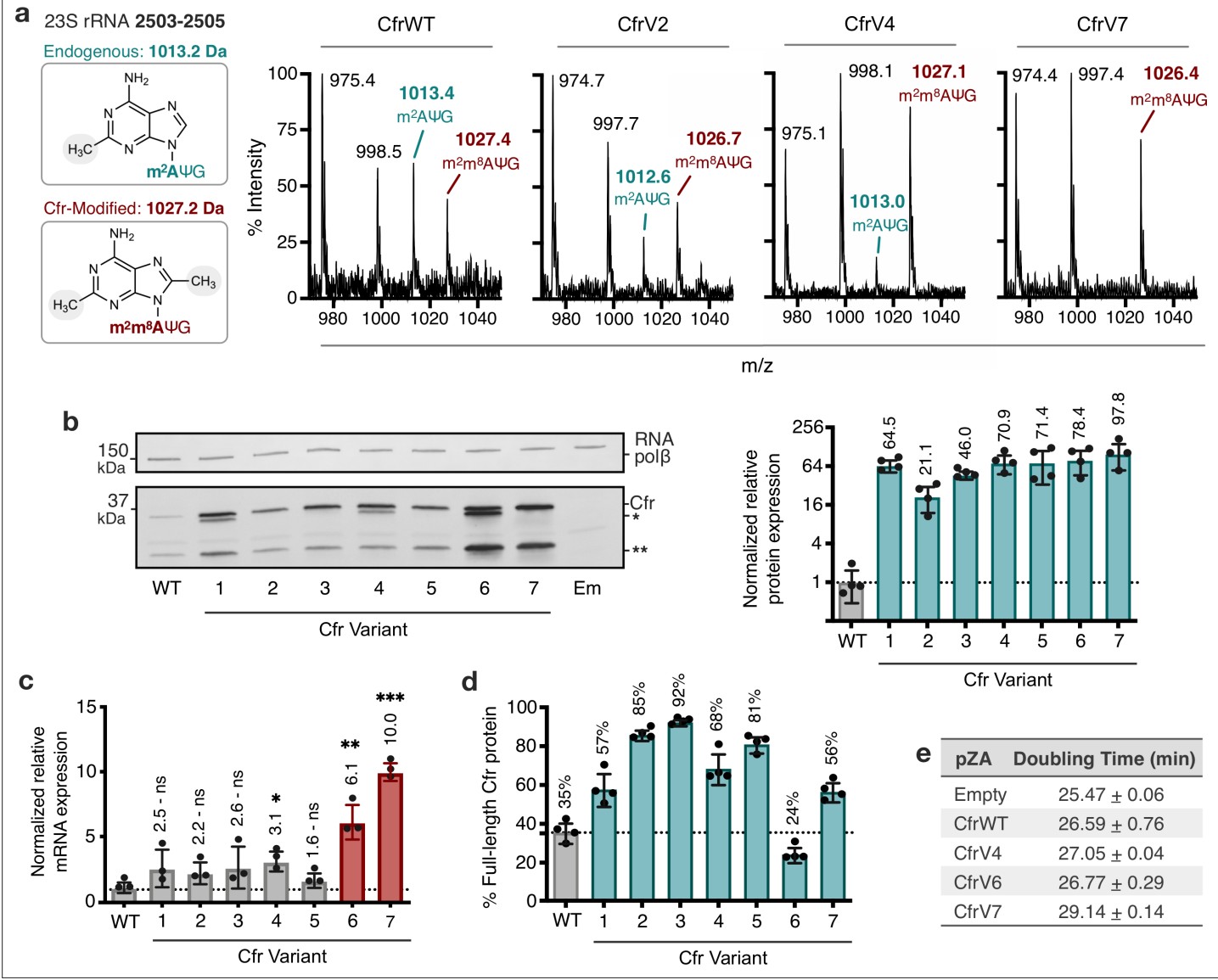

**Figure 2.** Cfr variants cause increased methylation of 23S rRNA at A2503, correlating with enhanced production of Cfr protein. (**a**) Endogenously modified (m²A2503) and Cfr-hypermodified (m²m⁸A2503) rRNA fragments correspond to m/z values of 1013 and 1027, respectively. MALDI-TOF mass spectra of 23S rRNA fragments isolated from *Escherichia coli* expressing CfrWT, and evolved Cfr variants V2, V4, and V7. Ψ is pseudouridine, m²A is 2-methyladenosine, is m²m⁸A is 2,8-dimethyladenosine. (**b**) Relative protein expression of full-length Cfr variants compared to full-length CfrWT detected by immunoblotting against a C-terminal FLAG tag and quantification of top Cfr bands. Signal was normalized to housekeeping protein RNA polymerase β-subunit. Data are presented as the average of four biological replicates with standard deviation on a log₂ axis. Asterisks denote N-terminally truncated versions of Cfr that do not contribute to resistance. Em = empty vector control. Original uncropped blot images are provided in *Figure 2—source data 1*. (**c**) Relative transcript levels for variants compared to CfrWT determined from three biological replicates with standard deviation. Statistical analysis was performed using a two-tailed t-test on log₂ transformed data. (**d**) Percentage of total Cfr expression attributed to the production of full-length Cfr protein, presented as the average of four biological replicates with standard deviation. (**e**) Doubling times for *E. coli* expressing empty plasmid, CfrWT, or Cfr variants were determined from three biological replicates with standard error. Numerical data and exact p values where relevant for panels (**b–d**) are provided in *Figure 2—source data 2*.

The online version of this article includes the following figure supplement(s) for figure 2:

**Source data 1.** Blot images.

**Source data 2.** Numerical and statistical data.

**Source data 3.** Numerical data.

**Source data 4.** Blot images.

*Figure 2 continued on next page*

eLife Research article    Biochemistry and Chemical Biology | Structural Biology and Molecular Biophysics

*Figure 2 continued*

**Source data 5.** Blot images.

**Source data 6.** Numerical data.

**Source data 7.** Numerical data.

**Figure supplement 1.** MALDI-TOF mass spectra of 23S rRNA fragments produced by oligo-protection and RNase $T_1$ digestion.

**Figure supplement 2.** In vitro characterization of CfrWT and CfrV4.

**Figure supplement 3.** N-terminally truncated Cfr products arise from internal Met translation start sites and do not contribute to resistance.

**Figure supplement 4.** Quantification of Cfr bands observed upon expression of Cfr variants.

In contrast, the evolved variants achieve ~50%–90% methylation of A2503, indicating that variants are more active than CfrWT in vivo.

The ability of evolved Cfr variants to achieve enhanced ribosome methylation in vivo could be attributed to enhanced enzymatic activity and/or higher levels of functional enzyme. To test the hypothesis that Cfr variants achieve higher turnover number, we anaerobically purified and reconstituted CfrWT and a representative evolved variant, CfrV4. We then evaluated the ability of CfrWT and CfrV4 to methylate a 23S rRNA fragment (2447–2625) in vitro by monitoring the incorporation of radioactivity from [$^3$H-methyl] $S$-adenosylmethionine (SAM) into RNA substrate under saturating conditions (*Bauerle et al., 2018*). However, no significant difference in $k_{cat}$ between CfrWT ($3.45 \times 10^{-2} \pm 3.2 \times 10^{-3}$ min$^{-1}$) and CfrV4 ($2.25 \times 10^{-2} \pm 1.3 \times 10^{-3}$ min$^{-1}$) was observed (*Figure 2—figure supplement 2*).

Given these findings, we hypothesized that the variants might alter protein levels. To monitor Cfr protein levels, we inserted a flexible linker followed by a C-terminal FLAG tag, which does not alter resistance (*Supplementary file 1*). Interestingly, immunoblotting against FLAG revealed that in addition to full-length Cfr, N-terminally truncated Cfr proteins are also produced (*Figure 2b*). The truncations result from translation initiation at internal methionines but do not contribute to resistance (*Figure 2—figure supplement 3*), indicating that they are non-functional enzymes unable to methylate A2503. The smaller molecular weight truncation is present in higher levels for all Cfr variants compared to CfrfWT (*Figure 2—figure supplement 4*). Interestingly the larger molecular weight truncation is present only in CfrV1/V4/V6 and is generated by the I26M mutation introduced during directed evolution. Quantification of resistance-causative, full-length Cfr proteins alone revealed that the evolved variants achieve ~20–100-fold higher steady-state protein levels than CfrWT (*Figure 2b*).

We measured transcript levels for all variants to assess the contribution of altered transcription to increased protein levels. For Cfr variants with promoter alterations, enhanced production of the Cfr transcript is a large contributor to Cfr protein expression, as CfrV6 and CfrV7 exhibit ~6-fold and ~10-fold enhancement in Cfr mRNA levels compared to CfrWT, respectively (*Figure 2c*). We also observe a ~2–3-fold increase in mRNA levels for CfrV1-5. Of note, increased Cfr transcript levels likely explain the higher expression of the larger molecular weight truncation observed for all variants (*Figure 2c*, *Figure 2—figure supplement 4*). Despite the observed increase in mRNA levels for CfrV1-5, this alone cannot explain the multi-fold improvement in protein expression and indicates that these variants also boost protein levels through a post-transcriptional process. This is further supported by the expression profiles for CfrV1-5, which are dominated by the full-length protein (*Figure 2d*). Interestingly, enhanced production of Cfr protein correlates with larger fitness defects in *E. coli*, with an increase in doubling time of ~4 min for CfrV7 compared to empty vector in the absence of antibiotics (*Figure 2e*).

## Promoter and second position mutations drive Cfr resistance

Given that the evolved variants achieve robust enhancement in Cfr expression we sought to elucidate the mechanism(s) by which this occurs. To evaluate the importance of promoter alterations, we generated a construct where the P$_{tet}$* promoter sequence from CfrV6 was inserted upstream of CfrWT ORF, herein referred to as P$_{tet}$*V6. The insertion of P$_{tet}$* alone was sufficient to elicit improvement in Cfr expression (*Figure 3a*). Furthermore, *E. coli* expressing P$_{tet}$*V6 resembled CfrV6 in its ability to survive in the presence of chloramphenicol (*Figure 3b*). Taken together, these results suggest the altered promoter drives expression and resistance for CfrV6.

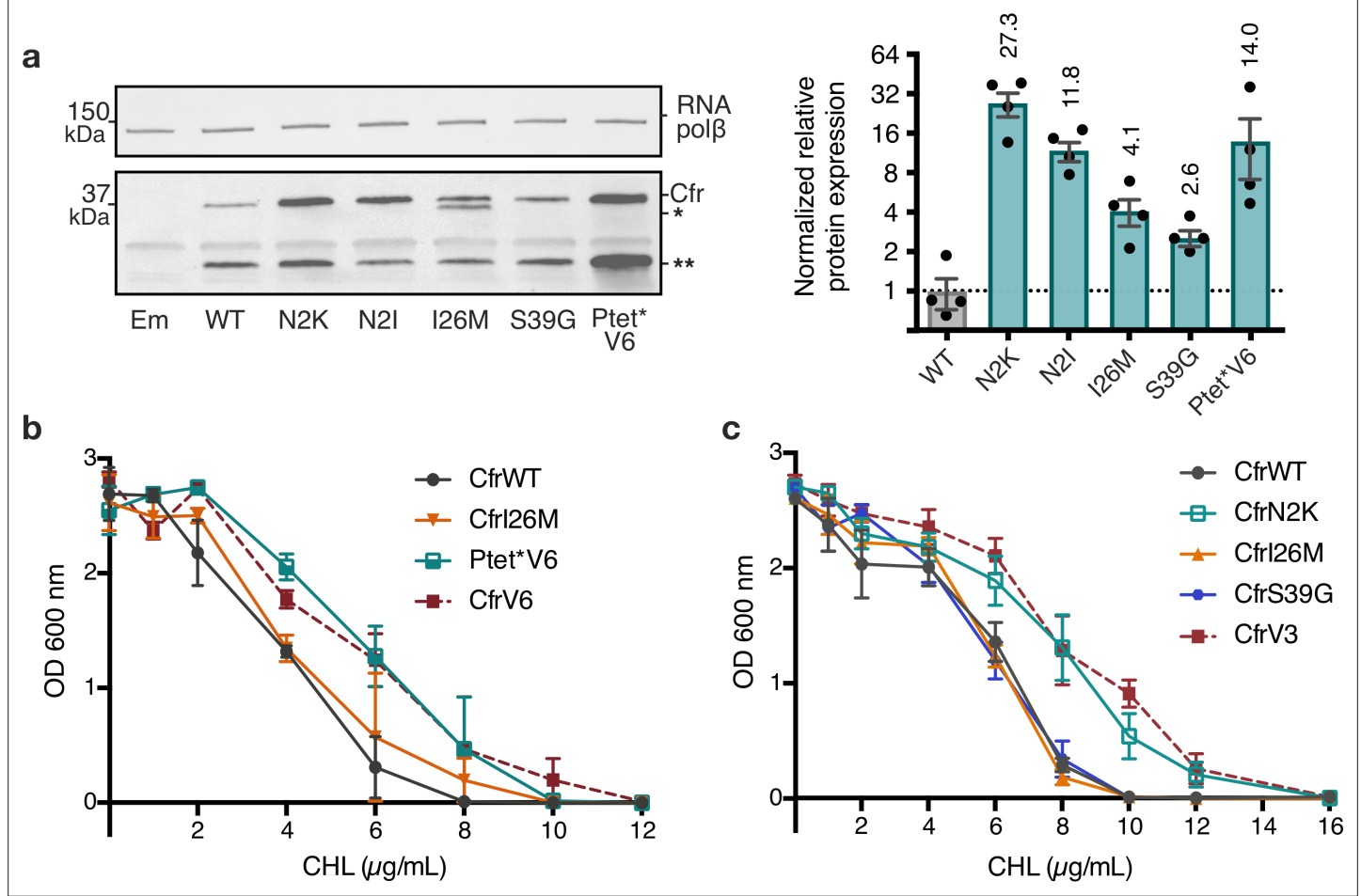

**Figure 3.** Mutations to the second amino acid and promoter are the largest contributors to Cfr expression and resistance. (**a**) Effect of Cfr mutations and promoter alteration on relative Cfr protein expression was assessed by immunoblotting against a C-terminal FLAG tag. Quantification was performed for full-length Cfr protein normalized to housekeeping protein RNA polymerase β-subunit. Data are presented as the average of four biological replicates with standard deviation on a $\log_2$ axis. Asterisks denote N-terminally truncated Cfr protein products that do not contribute to resistance and were not included in quantification. Em = empty vector control. Original uncropped blot images are shown in *Figure 3—source data 1*. (**b**) and (**c**) Dose-dependent growth inhibition of *Escherichia coli* expressing pZA-encoded CfrWT, CfrV6 (panel (**b**)), CfrV3 (panel (**c**)), and individual mutants that comprise these variants toward chloramphenicol (CHL) presented as an average of three biological replicates with standard error. Numerical data for all figure panels are provided in *Figure 3—source data 2*.

The online version of this article includes the following figure supplement(s) for figure 3:

**Source data 1.** Blot images.

**Source data 2.** Numerical data.

**Source data 3.** Numerical data.

**Source data 4.** Blot images.

**Source data 5.** Numerical data.

**Figure supplement 1.** Quantification of Cfr bands observed upon expression of Cfr Ptet*V6 or single mutations.

**Figure supplement 2.** Investigations into second position Cfr mutations N2I and N2K.

To investigate the contributions of mutations within the Cfr protein, we generated constructs containing Cfr mutations N2K/I, I26M, and S39G in isolation. Interestingly, we observe that mutations at the second position, N2K and N2I, display the largest enhancements in expression, ~27-fold and ~12-fold, respectively (*Figure 3a*). Similarly to the evolved variants, in addition to the full-length Cfr protein, we also observe expression of the truncation that results from initiation at M95 (*Figure 3—figure supplement 1*). The dominance of the second position mutants in driving antibiotic resistance is further manifested by *E. coli* expressing CfrN2K, but not I26M or S39G, exhibiting survival similar

to that of the triple mutant, CfrV3, in the presence of chloramphenicol (*Figure 3c*). Similarly, *E. coli* expressing CfrN2I also exhibits increased resistance to chloramphenicol when compared to the corresponding directed evolution variant, CfrV5, albeit weaker than CfrN2K (*Figure 3—figure supplement 2a*). Taken together, these results suggest that the second position mutations drive the robust expression and resistance observed for CfrV1-5. Of note, ribosome methylation by the produced Cfr does not impact the translation of CfrN2K, as this mutant and its corresponding catalytically inactive double mutant protein CfrN2K_C338A are similarly highly expressed (*Figure 3—figure supplement 2b-c*).

## Mutations impact Cfr translation and degradation

The Cfr coding mutations drive enhanced steady-state protein levels of Cfr protein through a post-transcriptional process. However, because levels at steady-state reflect the net effect of protein synthesis and degradation, we sought to evaluate how Cfr mutations impact both processes, especially since the nature of N-terminal amino acids and codons can greatly influence both translation and degradation in bacteria (*Gottesman, 2003*; *Bhattacharyya et al., 2018*; *Bentele et al., 2013*; *Tuller et al., 2010*; *Verma et al., 2019*; *Goodman et al., 2013*; *Boël et al., 2016*; *Looman et al., 1987*; *Stenström et al., 2001b*; *Stenström et al., 2001b*; *Stenström and Isaksson, 2002*; *Sato et al., 2001*).

To test the hypothesis that second position mutations enhance translation of mutants, we used polysome profiling to evaluate the relative abundance of Cfr mRNA in polysome fractions. Polysome profiles derived from 10% to 55% sucrose gradients appear similar across biological conditions, suggesting expression of CfrWT and its evolved mutants do not affect global translation (*Figure 4a–b*). CfrWT transcripts migrate with low polysomes (fractions 10 and 11) (*Figure 4c*). In contrast, CfrV4 transcripts are strongly shifted toward high polysomes (fractions 16 and 17), which indicate that CfrV4 mRNA is associated with a large quantity of ribosomes and is better translated than CfrWT (*Figure 4d*). Further support that CfrV4 is well-translated is the observation that CfrV4 mRNA co-migrates with mRNA of the well-translated housekeeping gene, *recA* (*Li et al., 2014*; *Figure 4—figure supplement 1*). At least in part, this is due to the N2K mutation which shifts transcripts to higher polysomes fractions (fractions 12 and 13) (*Figure 4c*). The *recA* control mRNA shows excellent reproducibility across biological samples, indicating that the observed shift of mutant Cfr transcripts toward higher polysomes is due to introduced mutations (*Figure 4b*). Taken together, these results suggest that enhanced translation is a cumulative effect of N2K and other ORF mutations obtained by directed evolution.

To further interrogate the role of second position mutations in Cfr translation, we determined the second codon identity for all sequenced variants from the final rounds of evolution (*Supplementary file 1*). Interestingly, all N2K mutations were encoded by an AAA codon, while AUU encoded all N2I mutations. In *E. coli*, the tRNA molecules that decode K(AAA) and I(AUU) are slightly more abundant than the wild-type N(AAU), accounting for 3.0% and 5.4% of the tRNA pool compared to 1.9%, respectively (*Dong et al., 1996*). To test if tRNA abundance and codon sequence contribute to enhanced translation, we evaluated the impact of synonymous codons on protein expression. Lysine codons AAA and AAG are decoded by the same tRNA$^{Lys}$ in *E. coli*. Interestingly, mutating CfrN2K from AAA to AAG, which increases G/C content, did not significantly impact expression (*Figure 4—figure supplement 2*). The isoleucine AUA codon is decoded by the low-abundant tRNA$^{Ile2}$ (*Del Tito et al., 1995*; *Nakamura et al., 2000*). Mutation of N2I from AUU to the AUA rare codon resulted in a ~2-fold decrease in Cfr expression, supporting tRNA abundance as a contributing factor (*Figure 4—figure supplement 2*).

To evaluate the impact of mutations introduced during directed evolution on protein half-life, we monitored changes in protein abundance over time after halting expression with rifampicin (*Figure 4e*, *Figure 4—figure supplement 3*). While CfrWT is rapidly degraded with a half-life of ~20 min, CfrN2K/I exhibit increased half-lives of ~60 min. These results suggest that mutation of the second amino acid to lysine or isoleucine contributes to improved steady-state expression both by enhancing translation and stability of Cfr in the cell. CfrS39G also exhibits an increased half-life of ~60 min. The half-life increase is the most pronounced for the I26M single point mutant and similar to that of the triple-mutant, CfrV3 (>100 min for both proteins). Together, these results suggest that evolved variants achieve higher expression through mutations that enhance translation and decrease the degradation of mutant Cfr proteins.

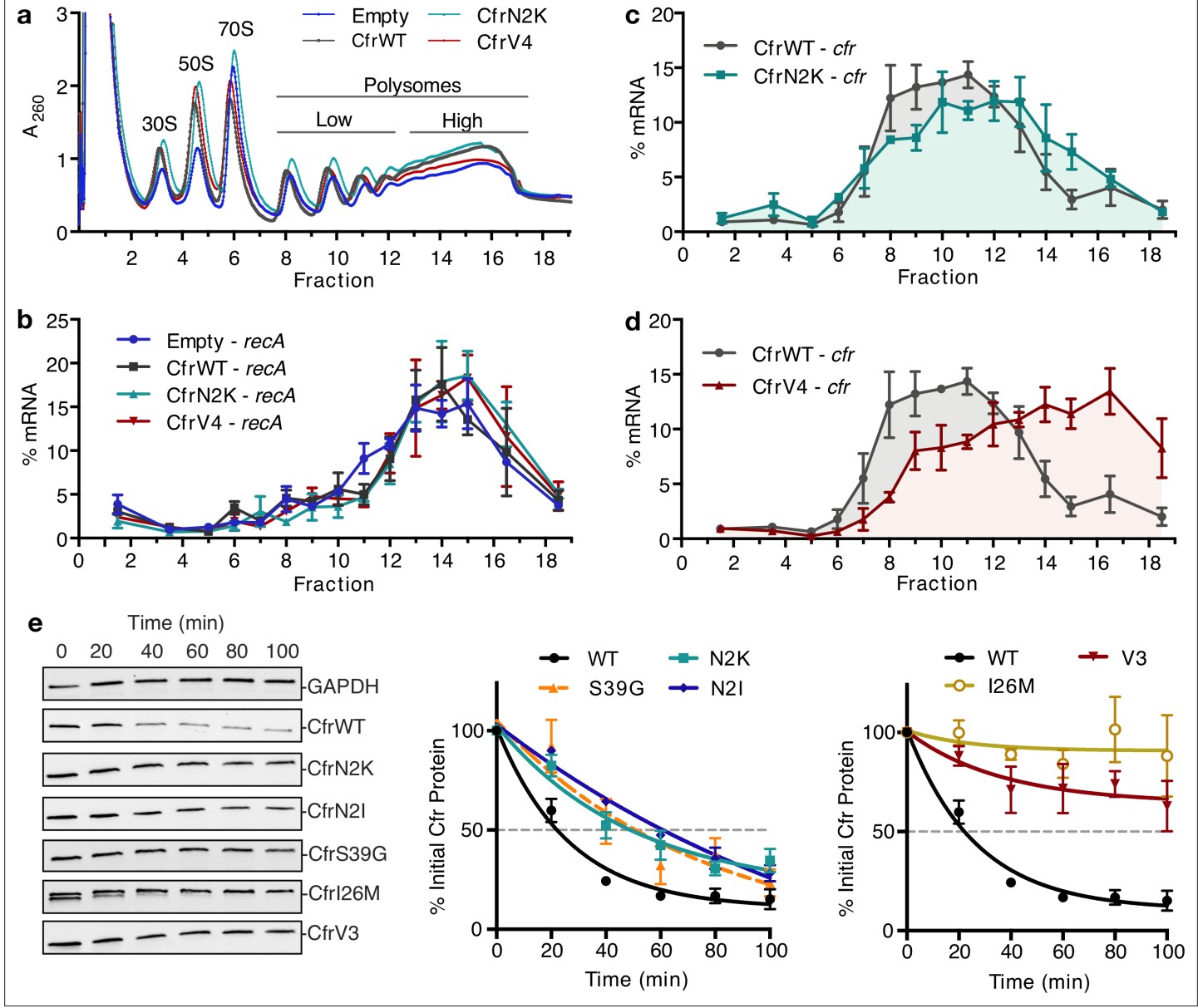

**Figure 4.** Directed evolution mutations impact Cfr translation and degradation. (**a**) Sucrose gradient fractionation of polysomes from *Escherichia coli* expressing empty vector or CfrWT/N2K/V4 denoting fractions corresponding to low- and high-density polysomes. (**b**) mRNA distribution of well-translated, housekeeping gene *recA* across polysome profiles. (**c**) mRNA distribution of Cfr transcripts expressing CfrWT or CfrN2K. (**d**) mRNA distribution of Cfr transcripts expressing CfrWT or CfrV4. For (**b**–**d**), transcript levels for each fraction were determined by RT-qPCR and normalized by a luciferase mRNA control spike-in. Values presented as the average of three biological replicates with standard error. (**e**) Protein degradation kinetics of CfrWT, single mutations CfrN2K/N2I/S39G/I26M, and evolved variant CfrV3 in *E. coli* after halting expression by rifampicin treatment. Percentage of Cfr protein remaining over time was determined by immunoblotting against C-terminal FLAG tag and presented as the average of three biological replicates with standard error. Original uncropped blot images are shown in *Figure 4—source data 1*. Numerical data for all figure panels are provided in *Figure 4—source data 2*.

The online version of this article includes the following figure supplement(s) for figure 4:

**Source data 1.** Blot images.

**Source data 2.** Numerical data.

**Source data 3.** Numerical data.

**Source data 4.** Blot images.

**Source data 5.** Numerical data.

*Figure 4 continued on next page*

*Figure 4 continued*

**Figure supplement 1.** Investigating translation of Cfr mutants.

**Figure supplement 2.** Impact of second codon identity on Cfr expression.

**Figure supplement 3.** Degradation of Cfr protein products.

**Figure supplement 4.** RNA secondary structure predictions of the sequence region proximal to the Cfr start codon.

**Figure supplement 5.** Protein sequence alignment of Cfr and Cfr homologs.

## Evolved Cfr enables understanding of the structural basis of resistance

Molecular understanding of Cfr-mediated resistance to antibiotics necessitates structural insights into methylated ribosomes. However, obtaining the structure of a Cfr-modified ribosome has been so far hampered by moderate methylation efficiency of *S. aureus* Cfr, a challenge that can be addressed by the improved methylation ability of directed evolution variants. Of all characterized evolved variants, CfrV7 achieves the highest levels of antibiotic resistance and methylation of rRNA, providing a unique tool for structural determination. Relative peak quantification of the MALDI spectra revealed that CfrV7 achieved near-stoichiometric (~90%) $m^8A2503$ methylation (*Figure 2—figure supplement 1*).

Ribosomes were purified from *E. coli* expressing CfrV7 to obtain a 2.2 Å cryo-EM structure of the Cfr-modified 50S ribosomal subunit (*Figure 5a*, *Table 1*, *Figure 5—figure supplement 1*). The high-resolution cryo-EM density map enabled modeling all known modified nucleotides including the novel C8 methylation of A2503 (*Figure 5b*). Furthermore, comparison of the Cfr-modified ribosome with the high-resolution cryo-EM structure of unmodified, wild-type *E. coli* ribosome we published previously (*Stojković et al., 2020*) allowed us to identify with high confidence any structural changes due to the presence of $m^8A2503$. Importantly, modification of A2503 by Cfr does not affect the conformation or position of the A2503 nucleotide. The adenine ring remains in the *syn*-conformation and places the newly installed C8-methyl group directly into the PTC to sterically obstruct antibiotic binding (*Figure 5c–d*).

Strikingly, beyond the addition of a single methyl group to the substrate nucleotide, the presence of $m^8A2503$ does not result in any additional structural changes to the PTC region of the ribosome (*Figure 5c*). Furthermore, the increased resistance provided by CfrV7 appears to be mediated specifically by improved methylation of A2503. No off-target activity of the evolved variant was observed as manual inspection did not reveal density that could correspond to additional C8-methyl adenosines within the high-resolution regions of the 50S ribosomal subunit. This result was cross-validated using our qPTxM tool (*Stojković et al., 2020*), which identified only A2503 and A556 as possible C8-methyl adenosines. Closer examination of A556 reveals it registered as a false positive (*Figure 5—figure supplement 2a-d*).

Contrary to previous reports, we do not observe changes to methylation of C2498, a distal PTC nucleotide whose endogenous 2′-O-ribose modification has previously been reported to be suppressed by Cfr methylation of A2503 and hypothesized to alter the PTC through long-range effects (*Kehrenberg et al., 2005*; *Jensen et al., 2009*; *Purta et al., 2009*). Although it is unclear what percentage of C2498 retains the native modification in our structure, we observe clear density for the methyl group and the nucleotide conformation is unaltered. The density for the methyl group is slightly off of the rotameric position, but the dropoff in density along the methyl bond matches the expected shape (*Figure 5—figure supplement 2e-g*). Taken together, the results do not indicate that conformational changes to C2498 are involved in Cfr-mediated resistance.

Although Cfr has been identified in animal-derived *E. coli* isolates (*Wang et al., 2012*; *Deng et al., 2014*; *Liu et al., 2017*; *Ma et al., 2021*), the resistance gene has primarily been identified clinically in staphylococcal organisms such as *S. aureus* (*Vester, 2018*). However, given the high sequence and structural conservation within the PTC region (*Figure 5—figure supplement 3*), structural impacts of the Cfr $m^8A2503$ modification within *E. coli* and *S. aureus* ribosomes are likely conserved.

Structural superposition of the Cfr-modified ribosome with ribosomes in complex with PhLOPS$_A$ antibiotics, hygromycin A, nucleoside analog A201A, and 16-membered macrolides enables direct identification of chemical moieties responsible for steric collision with $m^8A2503$ for these eight antibiotic drug classes (*Figure 5—figure supplement 4*, *Figure 5—figure supplement 5*). For example, overlay of a bacterial ribosome in complex with the pleuromutilin derivative tiamulin, the selection antibiotic used during directed evolution, reveals steric clashes between the C10 and C11 substituents

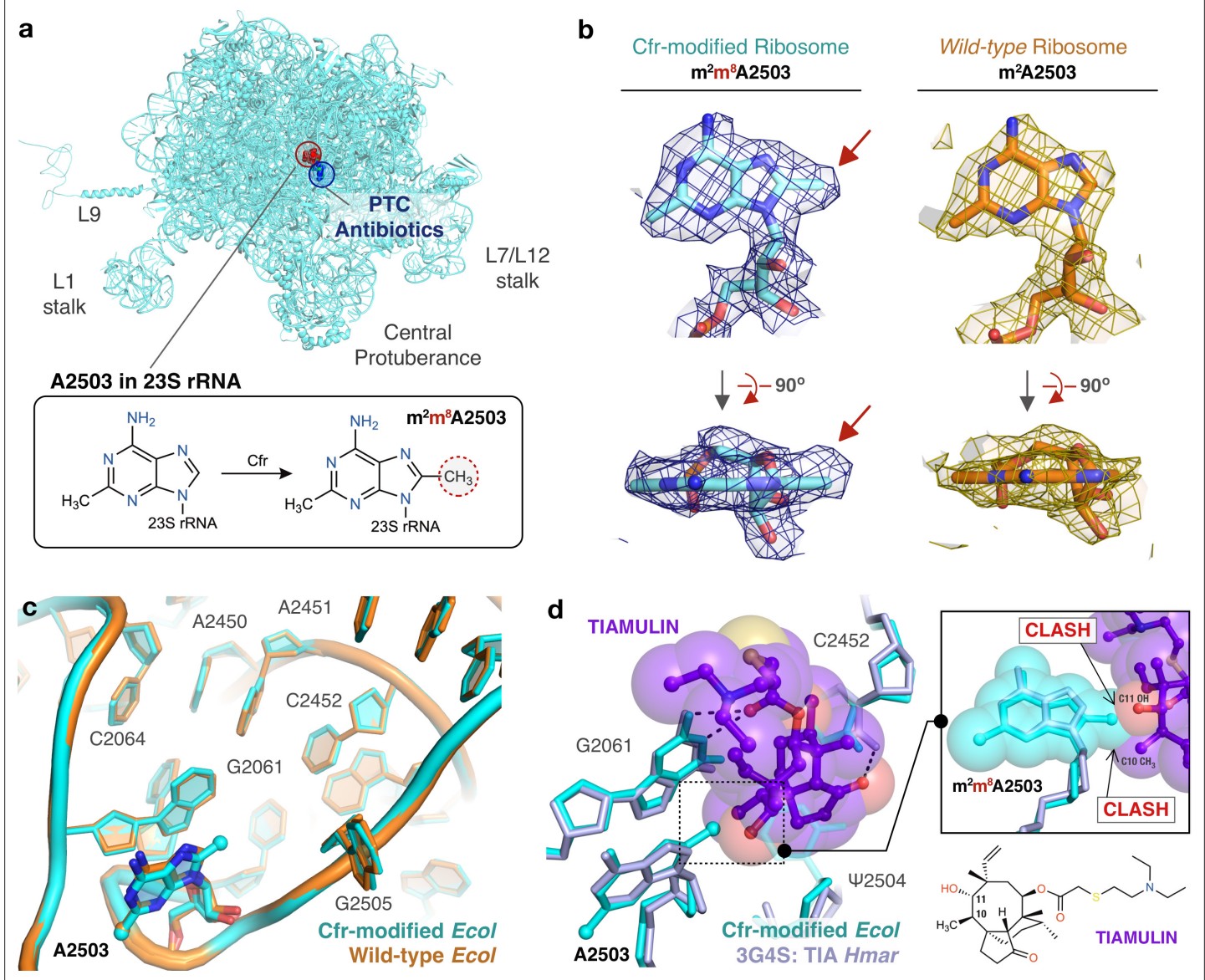

**Figure 5.** Near-stoichiometric ribosome methylation by CfrV7 enables structural understanding of Cfr-mediated resistance to antibiotics. (**a**) Cfr-modified 50S ribosomal subunit highlighting adenosine 2503 (A2503) within 23S rRNA and the binding site of PTC-targeting antibiotics. Cfr methylates A2503 at the C8 carbon to produce $m^2m^8A2503$. (**b**) Cryo-EM density maps of adenosine 2503 in 23S rRNA contoured to 3σ. Cfr-modified ($m^2m^8A2503$) in cyan. Wild-type ($m^2A2503$) in orange; PDB 6PJ6. (**c**) Close-up view of 23S rRNA nucleotides in the 50S ribosomal subunit. Cfr-modified *Escherichia coli* ribosome in cyan. Wild-type *E. coli* ribosome in orange; PDB 6PJ6. (**d**) Structural overlay of Cfr-modified *E. coli* ribosome (cyan) and *Haloarcula marismortui* 50S ribosome in complex with pleuromutilin antibiotic tiamulin (purple, PDB 3G4S) highlighting steric clashes between $m^8A2503$ and the antibiotic. EM, electron microscopy.

The online version of this article includes the following figure supplement(s) for figure 5:

**Figure supplement 1.** Cryo-EM data collection and processing of the Cfr-modified ribosome.

**Figure supplement 2.** Cross-validation of methylations on C8 of A2503 and 2'O of C2498 from the cryo-EM density map.

**Figure supplement 3.** Overlay of Cfr-modified *Escherichia coli* ribosome and WT *Staphylococcus aureus* ribosome.

**Figure supplement 4.** Molecular basis of Cfr-mediated resistance to PhLOPS_A antibiotics.

**Figure supplement 5.** Molecular basis of Cfr-mediated resistance to hygromycin A, A201A, and 16-membered macrolide antibiotics.

**Table 1.** Cryo-EM data collection, refinement, and validation statistics.

| Data collection and processing | |
| --- | --- |
| Electron microscope | Krios |
| Magnification | 29,000 |
| Number of micrographs | 2055 |
| Number of particles picked from good micrographs | 162,713 |
| Number of particles used in final reconstruction | 141,549 |
| Pixel size (Å) | 0.822 |
| Defocus range (μm) | –0.2 to –1.5 |
| Voltage (kV) | 300 |
| Electron dose (e-/Å2) | 80 |
| **Map refinement** | |
| Model resolution (Å) | 2.2 |
| FSC threshold | 0.143 |
| Model resolution range (Å) | 2.2–20 |
| Map sharpening B-factor (Å2) | –55.86 |
| **Refinement and model statistics** | |
| Clashscore, all atoms | 2.23 |
| Protein geometry | |
| MolProbity score | 1.29 |
| Rotamer outliers (%) | 0.92 |
| Cβ deviations > 0.25 Å (%) | 0.32 |
| Ramachandran (%) | |
| - Favored | 95.79 |
| - Allowed | 4.01 |
| - Outliers | 0.2 |
| Deviations from ideal geometry | |
| - Bonds (%) | 0.03 |
| - Angles (%) | 0.08 |
| Nucleic acid geometry | |
| Probably wrong sugar puckers (%) | 0.84 |
| Bad backbone conformations (%) | 12.86 |
| Bad bonds (%) | 0.07 |
| Bad angles (%) | 0.08 |

of the antibiotic with the Cfr-introduced methyl group (*Figure 5d*). The pleuromutilin class of antibiotics has recently regained interest for their applications as antimicrobial agents in humans but existing molecules remain ineffective against pathogens with Cfr (*Goethe et al., 2019*). Given recent synthetic advances that enable more extensive modification of the pleuromutilin scaffold (*Murphy et al., 2017*; *Farney et al., 2018*), the structural insights we obtained will inform the design of next-generation antibiotics that can overcome Cfr-mediated resistance.

## Discussion

By relying on directed evolution under antibiotic selection, we identified strategies that increase the ability of a multi-antibiotic resistance determinant Cfr to cause resistance. Enhanced resistance is associated with improved in vivo methylation of rRNA at the C8 position of A2503. The positive correlation between extent of rRNA modification and resistance aligns with previous studies that investigated linezolid resistance caused by mutation of rRNA, where the severity of linezolid resistance was proportional to the number of 23S rRNA alleles harboring the resistance mutation (*Besier et al., 2008*; *Ebihara et al., 2014*; *Lobritz et al., 2003*). While alteration of the antibiotic binding site through mutations and enzymatic modification of 23S rRNA are functionally distinct, dependence on the extent of rRNA modification provides parallels between the two mechanisms. Although Cfr-mediated methylation is an enzymatic process, the ability of Cfr to confer resistance is restricted by ribosome assembly. Since the A2503 is only accessible to Cfr prior to incorporation of 23S rRNA into the large ribosomal subunit (*Yan et al., 2010*), the extent of resistance correlates with the ability of the enzyme to methylate 23S rRNA prior to its incorporation into the 50S subunit. The results of our evolution experiment indicate that increasing the intracellular concentrations of Cfr, rather than improving catalysis of an enzyme with a complex radical mechanism (*Yan and Fujimori, 2011*; *McCusker et al., 2012*; *Grove et al., 2011*; *Bauerle et al., 2018*) is the preferred strategy to increase the proportion of ribosomes with the protective m8A2503 modification.

Investigations into expression levels of CfrWT and its respective mutants revealed that, in addition to full-length protein, a smaller Cfr isoform of ~30 kDa is also produced (*Figures 2b and 3a*). The truncated product is observed when the expression is driven by both the non-native $P_{tet}$ and native $P_{cfr}$ promoter (*Figure 6*, *Figure 6—figure supplement 1*). The smaller product likely results from translation initiation at an internal start codon, as mutation of Met at position 95 abolishes its production. The sequence upstream of M95 is A-rich,

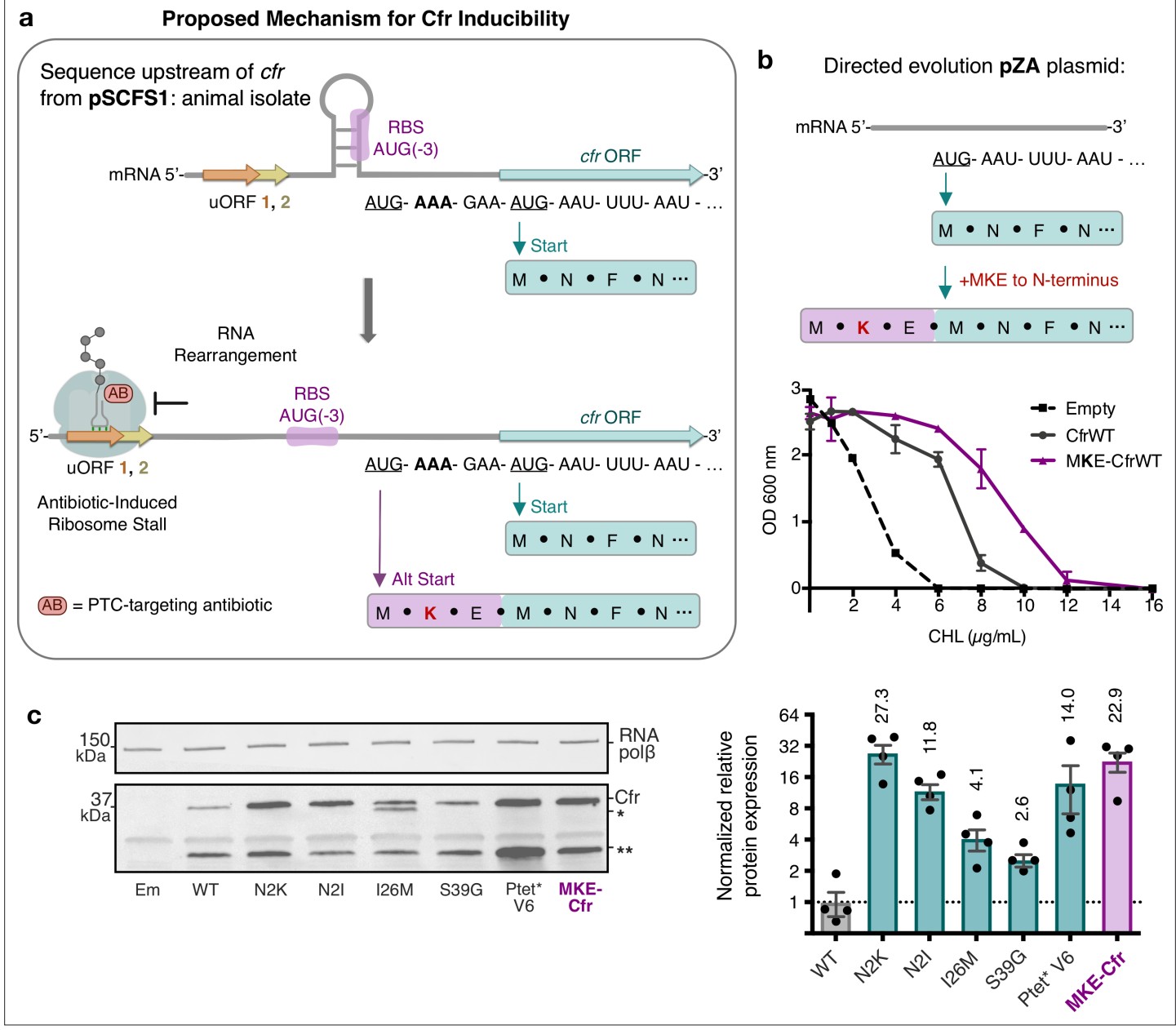

**Figure 6.** Start codon selection as a proposed mechanism of Cfr inducibility. (**a**) Sequence upstream of *cfr* from pSCFS1 (Accession: AJ579365) resistance plasmid from an animal-derived *Staphylococcus sciuri* isolate. The upstream region contains two overlapping upstream ORFs (uORFs) followed by an RNA structural element. Proposed antibiotic-induced ribosome stalling at uORF1/2 and RNA rearrangement could reveal the occluded RBS, allowing translation to initiate at AUG(–3), adding an MKE polypeptide to the N-terminus of Cfr. (**b**) Addition of an N-terminal MKE peptide to Cfr in the context of the pZA plasmid where expression is controlled by the non-native tetracycline-inducible promoter, $P_{tet}$. Growth inhibition of *Escherichia coli* with pZA-encoded MKE-Cfr in the presence of CHL was determined from two biological replicates with standard error. (**c**) Relative protein expression of full-length MKE-Cfr compared to full-length CfrWT detected by immunoblotting against a C-terminal FLAG tag and quantification of top Cfr bands. Signal was normalized to housekeeping protein RNA polymerase β-subunit. Data are presented as the average of four biological replicates with standard deviation on a $\log_2$ axis. Em = empty vector control. Numerical data for figure panels are provided in *Figure 6—source data 1*.

The online version of this article includes the following figure supplement(s) for figure 6:

**Source data 1.** Numerical data.

**Source data 2.** Blot images.

**Source data 3.** Numerical data.

**Figure supplement 1.** Investigations into Cfr translation start sites in the pSCFS1 sequence context.

which has been demonstrated to promote translation initiation (*Saito et al., 2020*). However, why an internal region of the Cfr ORF would be recognized as an initiation signal is unclear. Truncation of the first 38 residues of Cfr would eliminate a significant portion of the protein, including the N-terminal accessory domain which is likely involved in substrate recognition (*Boal et al., 2011*; *Schwalm et al., 2016*). Elimination of this domain provides rationale for why the smaller Cfr isoform does not contribute to resistance, as the protein would likely exhibit perturbed binding to rRNA. Thus, while the truncated product does not contribute to resistance, the potential function of the smaller protein remains elusive and requires further study.

The evolved variants improve the expression of resistance-causative, full-length Cfr using two mechanisms. Improved Cfr expression for CfrV6/7 is driven by increased transcription (*Figure 2c*) due to alterations to the $P_{tet}$ promoter likely introduced by primer slippage during the EP-PCR step of directed evolution. CfrV6 contains a full duplication of $P_{tet}$, providing two sites for transcription initiation, likely responsible for enhanced *cfr* transcript levels. Interestingly, this result parallels a clinical instance of high Cfr resistance discovered in an *S. epidermidis* isolate where transcription of *cfr* was driven by two promoters (*LaMarre et al., 2013*) and highlights transcriptional regulation as an important mechanism for modulating the in vivo activity of Cfr.

Improved expression for evolved variants CfrV1-5 is mediated by mutations that improve both translational efficiency and protein stability in vivo. Of the tested mutations, I26M provides the largest improvement in stability. Of note, the N-terminally truncated Cfr derived from translation initiation at I26M is rapidly degraded, as no detectable protein is observed after 60 min (*Figure 4e*). However, these results indicate that the costly production and clearance of this nonfunctional protein is offset by the improved cellular stability of the full-length Cfr carrying the I26M mutation. We also observe modest improvements in protein stability with N2K/I mutants (*Figure 4e*). In bacteria, the identity of N-terminal residues are important determinants of degradation through N-degron pathways (*Dougan et al., 2012*; *Tobias et al., 1991*). During protein synthesis, the N-terminus is co-translationally processed by two enzymes, peptide deformylase to remove the formyl group from Met (fM) and methionine aminopeptidase (*Koubek, 2021*). Based on previous biochemical work, it is unlikely that CfrWT and CfrN2K/I would have different N-terminal processing, since fMN… and fMK/I… are likely to be efficiently de-formylated (*Ragusa et al., 1999*) and resistant to methionine excision (*Hirel et al., 1989*; *Frottin et al., 2006*; *Xiao et al., 2010*). Although the precise mechanism by which N2K/I improves Cfr stability remains elusive, these mutations may alter recognition by other enzymes important for degradation, such as endopeptidases or L/F-tRNA-protein transferase (*Izert et al., 2021*; *Ottofuelling et al., 2021*).

Of the mutations investigated, N2K is the largest contributor to enhanced Cfr expression and resistance. Although N2K contributes to cellular stability, our results suggest that improved Cfr translation is the dominant role of this mutation (*Figure 4c*). The effect of N-terminal residues (and thus codons near the start site) on early stages of translation has been well documented. Previous work has demonstrated that minimal secondary structure near the start codon (*Kudla et al., 2009*; *Goodman et al., 2013*; *Bentele et al., 2013*; *Boël et al., 2016*; *Bhattacharyya et al., 2018*) and presence of A/U-rich elements downstream of the translation start site (*Cifuentes-Goches et al., 2019*; *Saito et al., 2020*) are important factors for efficient translation initiation. RNA secondary structure predictions of the region proximal to the start codon suggest that the N2K mutation (AAU to AAA) could disrupt base pairing between the N2 (AAU) codon and the downstream T7 (ACA) codon (*Figure 4—figure supplement 4*). However, the base pair between the second and seventh codon is predicted to be retained when N2K is encoded by the AAG isocodon (*Figure 4—figure supplement 4*), which was also able to increase Cfr protein levels (*Figure 4—figure supplement 2*) and suggests that alternative mechanisms may be responsible for improved translation. While initiation is a major rate-limiting step in protein synthesis, rates of elongation have also been demonstrated to impact translation efficiency, with several proposed models on how the interconnected factors of codon bias, mRNA structure/sequence, and interactions between the ribosome and nascent chain are involved in modulating protein synthesis (*Rodnina, 2016*; *Choi et al., 2018*; *Samatova et al., 2020*). For example, recent work investigating the role of codons 3–5 identified that both mRNA sequence and amino acid composition are key determinants of proper elongation at the N-terminus (*Verma et al., 2019*). Although the mechanism is poorly understood, previous studies have discovered that presence of an AAA lysine codon after the start site is associated with improved translation efficiency in certain

contexts (*Looman et al., 1987*; *Stenström et al., 2001a*; *Stenström et al., 2001b*; *Stenström and Isaksson, 2002*; *Sato et al., 2001*). Our results indicate that the effect of N2K on early steps of translation elongation may be mediated, at least in part, by tRNA abundance, but the exact impact of Lys2 on translation requires further study. Interestingly, the observed internal translation start sites (I26M and M95) that are responsible for producing Cfr truncations (*Figure 2b*, *Figure 2—figure supplement 3*) contain a lysine immediately after methionine, further highlighting the putative role for lysine codons in early steps of translation.

To date, only a few *S. aureus* Cfr variants have been reported and no mutations matching those obtained from directed evolution have been found in clinical isolates. However, enhanced expression through positioning of Lys as the second amino acid of Cfr can be recapitulated by accessing an upstream translational start site found in a native sequence context of *cfr* (*Figure 6*). In the specific case of the pSCFS1 resistance plasmid, the sequence upstream of the annotated start codon, which we validated as the start site under the experimental conditions tested (*Figure 6—figure supplement 1*), contains regulatory elements that have been proposed to modulate Cfr expression (*Schwarz et al., 2000*; *Kehrenberg et al., 2007*). It is plausible that in response to antibiotics, the upstream start codon is used to add three amino acids (MKE) to the N-terminus of Cfr and thus placement of a lysine (K) at position 2 of the newly expressed protein, analogous to the N2K mutation. Although start codon selection requires further investigation, N-terminal addition of MKE to Cfr expressed under non-native $P_{tet}$ promoter phenocopies the N2K directed evolution mutation, resulting in increased expression and resistance compared to CfrWT (*Figure 6*). Since our assessment of the evolved variants indicates that an increase in Cfr expression is accompanied by a decrease in fitness (*Figure 2e*), start site selection in response to antibiotic pressure would mitigate detrimental impact on fitness while enabling higher resistance when acutely needed.

Interestingly, mutations obtained through directed evolution have been observed in Cfr homologs that share less than 80% sequence identity with Cfr. Methionine (M) at position 26 is observed for the functionally characterized Cfr homologs Cfr(B) (*Deshpande et al., 2015*; *Marín et al., 2015*; *Hansen and Vester, 2015*) and Cfr(D) (*Pang et al., 2020*), which have been recovered from human-derived isolates and share 74% and 64% amino acid identity with Cfr, respectively (*Schwarz et al., 2021*; *Figure 4—figure supplement 5*). We also observe lysine (K) at position 2, methionine (M) at position 26, and glycine (G) at position 39, akin to N2K, I26M, and S39G mutations, for a number of Cfr homologs that clade with functional Cfr or Cfr-like genes (*Stojković et al., 2019*). While the precise roles of these residues within less well-characterized and more distantly related Cfr proteins requires further study, these observations indicate that directed evolution accessed sequence space that is already being exploited by proteins that are, or are hypothesized to be, functional Cfr resistance enzymes.

In addition to identifying mechanisms that increase Cfr-mediated resistance, directed evolution of Cfr also provided an indispensable reagent that enabled structural determination of the Cfr-modified ribosome. The high-resolution cryo-EM structure revealed that broad resistance is due to steric effects of the judiciously positioned methyl group within the shared binding site of PTC-targeting antibiotics. Lack of notable changes in position or orientation of A2503 or surrounding PTC nucleotides upon Cfr methylation suggests that the resulting modification does not obstruct the translation capabilities of the ribosome. This absence of PTC disruption is consistent with the observation that the fitness cost of Cfr acquisition is not due to ribosome modification, but rather results from expression of the exogenous protein (*LaMarre et al., 2011*). Importantly, overlay with existing structures containing PTC-targeting antibiotics provides direct visualization of chemical moieties that are sterically impacted by $m^8A2503$ and will inform the design of antibiotic derivatives that can overcome resistance mediated by Cfr.

## Materials and methods

### Key resources table

| Reagent type (species) or resource | Designation | Source or reference | Identifiers | Additional information |
|---|---|---|---|---|
| Strain, strain background (*E. coli*) | ER2267 | Tawfik lab stock | | |

*Continued on next page*

*Continued*

| Reagent type (species) or resource | Designation | Source or reference | Identifiers | Additional information |
|---|---|---|---|---|
| Strain, strain background (*E. coli*) | BW25113 | Keio collection | | |
| Strain, strain background (*E. coli*) | BW25113 *acrB::Kan* | Keio collection | | |
| Strain, strain background (*E. coli*) | Rosetta2 BL21(DE3) pLysS | Novagen | | |
| Strain, strain background (*E. coli*) | MRE600 | ATCC | | |
| Gene (*Staphylococcus aureus*) | *cfr* gene | | Accession: EF450709.1 | |
| Gene (*Staphylococcus sciuri*) | Sequence upstream *cfr* gene | Genscript | Accession: AJ579365 | |
| Recombinant DNA reagent | pZA (plasmid) | *Wellner et al., 2013*; *Stojković et al., 2016* | | |
| Recombinant DNA reagent | pET28a (plasmid) | Fujimori lab stock | | |
| Recombinant DNA reagent | pKK3535 (plasmid) | Fujimori lab stock | | |
| Chemical compound, drug | Anhydrotetracycline hydrochloride | Sigma-Aldrich | 37919 | |
| Chemical compound, drug | Tiamulin | Wako Chemicals | 328-34002 | |
| Chemical compound, drug | Clindamycin | TCI America | C2256 | |
| Chemical compound, drug | Chloramphenicol | Allstar Scientific | 480-045 | |
| Chemical compound, drug | Linezolid | Acros | 460592500 | |
| Chemical compound, drug | Hygromycin A | Dr. Kim Lewis | | |
| Chemical compound, drug | Trimethoprim | Sigma-Aldrich | 92131 | |
| Chemical compound, drug | [3H-methyl] S-adenosylmethionine | PerkinElmer | NET 155 H001MC | |
| Chemical compound, drug | Rifampicin | Sigma-Aldrich | R3501 | |
| Antibody | Anti-FLAG (mouse monoclonal) | Sigma-Aldrich | F3165 | 1:2000 |
| Antibody | Anti-RNA polymerase beta (rabbit monoclonal) | Abcam | ab191598 | 1:2000 |
| Antibody | Anti-rabbit IgG cross-absorbed DyLight 680 (goat polyclonal) | Thermo Fisher Scientific | 35567 | 1:10,000 |
| Antibody | Anti-mouse IgG cross-absorbed IRDye 800CW (goat polyclonal) | Abcam | ab216772 | 1:10,000 |
| Antibody | Anti-GAPDH (mouse monoclonal) | Abcam | ab125247 | 1:2000 |
| Sequence-based reagent | cfr | Integrated DNA Technologies | | For RT-qPCR. The sequence of this oligonucleotide can be found in *Supplementary file 1* |
| Sequence-based reagent | recA | Integrated DNA Technologies | | For RT-qPCR. The sequence of this oligonucleotide can be found in *Supplementary file 1* |
| Sequence-based reagent | luc | Integrated DNA Technologies | | For RT-qPCR. The sequence of this oligonucleotide can be found in *Supplementary file 1* |

*Continued on next page*

*Continued*

| Reagent type (species) or resource | Designation | Source or reference | Identifiers | Additional information |
|---|---|---|---|---|
| Sequence-based reagent | Luciferase control RNA | Promega | L456A | Control RNA-spike in for polysome analysis |
| Commercial assay or kit | RNAprotect Bacteria Reagent | QIAGEN | 76506 | |
| Commercial assay or kit | RNeasy Mini Kit | QIAGEN | 74104 | |
| Commercial assay or kit | iScript cDNA Synthesis Kit | Bio-Rad | 170-8891 | |
| Commercial assay or kit | SsoAdvanced Universal SYBR Green Supermix | Bio-Rad | 172-5721 | |
| Peptide, recombinant protein | RNaseT1 | Thermo Fisher Scientific | EN0541 | |
| Peptide, recombinant protein | RQ1 RNase-free DNase I | Promega | M610A | |
| Software, algorithm | Image Lab Software | Bio-Rad | | |
| Software, algorithm | MotionCor2 | *Zheng et al., 2017* | | |
| Software, algorithm | GCTF | *Zhang, 2016* | | |
| Software, algorithm | CryoSPARC v2.0 | *Punjani et al., 2017* | | |
| Software, algorithm | Relion 3.0 | *Scheres, 2012* | | |
| Software, algorithm | cisTEM | *Grant et al., 2018* | | |
| Software, algorithm | Sharpen3D | *Grant et al., 2018* | | |
| Software, algorithm | PHENIX | *Adams et al., 2010* | | |
| Software, algorithm | eLBOW | *Moriarty et al., 2009* | | |
| Software, algorithm | MolProbity | *Chen et al., 2010* | | |
| Software, algorithm | Pymol Molecular Graphics System | Schrödinger, LLC | | Version 2.4.1 |

## *E. coli* strains and plasmids

*E. coli* ER2267 expressing Cfr from a pZA vector (*Wellner et al., 2013*; *Stojković et al., 2016*) was used in directed evolution experiments. Antibiotic resistance, fitness, in vivo RNA methylation, and protein/transcript expression, polysome analysis, and protein degradation experiments were conducted with *E. coli* BW25113 expressing Cfr protein from a pZA vector under the $P_{tet}$ promoter (or $P_{cfr}$ promoter where noted). *E. coli* BW25113 *acrB::kan*, where the efflux pump *acrB* was replaced with a kanamycin cassette, was used for antibiotic susceptibility testing of the oxazolidinone antibiotic linezolid and hygromycin A. For experiments for which tagless versions of evolved Cfr variants were used, comparisons were made to the wild-type Cfr protein to which the original C-terminal His tag had been removed. *E. coli* Rosetta2 BL21(DE3) pLysS was used for overexpression of N-His$_6$-SUMO-tagged Cfrs from a pET28a vector. *E. coli* MRE600 was used for the preparation of Cfr-modified ribosomes for structural studies.

## Cfr mutagenesis and selection scheme

The wild-type *cfr* gene (accession: EF450709.1) with a C-terminal His$_6$-tag, or pooled *cfr* genes from the previous round of evolution, were randomly mutagenized by error-prone polymerase chain reaction as described previously (*Stojković et al., 2016*). The mutagenized *cfr* gene pool was then recloned into a pZA vector and transformed into *E. coli* ER2267. The frequency of mutations was determined by sequencing randomly selected library variants and was on average 1–3 mutations per gene. *E. coli* transformants were then subjected to selection by plating cells on LB agar containing tiamulin (Wako Chemicals USA), in addition to 100 µg/ml ampicillin for plasmid maintenance and 20 ng/ml anhydrotetracycline (AHT, Sigma-Aldrich) for induction of Cfr expression. For each round of evolution, the *E. coli* transformants were divided equally and plated on 4–5 plates of LB agar containing different concentrations of tiamulin and grown at 37 °C for up to 48 hr. The tiamulin concentration

was increased in 50–100 µg/ml increments. For example, in the first round of evolution, the transformation was plated on the 150, 200, 250, and 300 µg/ml tiamulin plates, in the last round, we selected on 250, 350, 450, and 550 µg/ml tiamulin plates. About 2 ml was plated on tiamulin deficient plates in order to determine transformation efficiency. In general, colonies isolated from tiamulin plates in which the ≤10% of the transformants grew were taken for the next round. After five rounds of mutagenesis and selection, two rounds of enrichment (selection without mutagenesis) using high tiamulin concentrations (400–1500 µg/ml) was conducted. After each round of selection or enrichment, 5–10 randomly selected colonies were sequenced from each plate.

## Determination of antibiotic resistance

Antibiotic resistance experiments by broth microdilution followed established protocols (*Wiegand et al., 2008*). In brief, 2 ml of LB media with selection antibiotic was inoculated with a freshly transformed colony containing either empty plasmid, CfrWT, or Cfr mutants. Cultures were grown at 37°C with shaking for approximately 2.5 hr. After measuring the $OD_{600}$ value, cultures were diluted to $10^6$ cells and 50 µl of this dilution was dispensed into 96-well plates containing 50 µl of LB media with antibiotic of interest, ampicillin (100 µg/ml), and AHT (30 ng/ml). Antibiotic resistance of evolved Cfr variants was evaluated using a twofold serial dilution of antibiotic with the following concentration ranges: tiamulin (50–6400 µg/ml, Wako Chemicals); clindamycin (50–6400 µg/ml, TCI America), chloramphenicol (0.5–64 µg/ml, AllStar Scientific), linezolid (1–256 µg/ml, Acros), hygromycin A (2–1024 µg/ml, gifted from Dr. Kim Lewis), and trimethoprim (0.125–0.2 µg/ml, Sigma-Aldrich). Chloramphenicol resistance of single Cfr mutations was evaluated using concentrations of 1, 2–12 µg/ml (in 2 µg/ml step increments), followed by 16–64 µg/ml (twofold dilution). The minimum inhibitory concentration (MIC) required to inhibit visible bacterial growth was determined after incubating plates at 37°C with shaking for 18 hr. Plate $OD_{600}$ values were also recorded with a microtiter plate reader (SpectraMax M5, Molecular Devices). Antibiotic resistance determination on LB agar plates was conducted as described previously (*Stojković et al., 2016*; *Wiegand et al., 2008*). In brief, 3 µl of $10^8$, $10^6$, and $10^4$ dilutions *E. coli* harboring Cfr were spotted on LB agar plates containing various concentrations of tiamulin. LB agar plates also contained ampicillin (100 µg/ml) and AHT (30 ng/ml). LB agar plates were incubated at 37°C for 24–48 hr.

## Oligo-protection of rRNA and MALDI-TOF analysis

*E. coli* expressing empty plasmid or Cfr were grown at 37°C to an $OD_{600}$ of 0.4–0.6 with shaking by diluting an overnight culture 1:100 into LB media containing ampicillin (100 µg/ml) and AHT inducer (30 ng/ml). Total RNA purification, oligo-protection of the 23S rRNA fragment C2480-C2520, and RNaseT1 digestion were performed as described previously (*Stojković and Fujimori, 2015*; *Andersen et al., 2004*). Mass spectra were acquired in positive ion, reflectron mode on an AXIMA Performance MALDI TOF/TOF Mass Spectrometer (Shimadzu). Relative peak intensity values were calculated using the Shimadzu Biotech MALDI-MS software.

## Expression and purification of Cfr

CfrWT and CfrV4 were expressed, purified, and reconstituted using modified published protocols (*Yan et al., 2010*; *Stojković and Fujimori, 2015*). In brief, N-His$_6$-SUMO-tagged CfrWT/V4 was overexpressed in minimal media conditions with 800 µM IPTG and 1,10-phenanthroline to obtain enzyme lacking a [4Fe-4S] iron-sulfur cluster. Minimal media also contained selection antibiotics kanamycin (50 µg/ml) and chloramphenicol (34 µg/ml). All purification steps were performed in a glovebox (MBraun, oxygen content below 1.8 ppm) that was cooled to 10°C. Cfr was purified by Talon chromatography (Clontech) from clarified lysates. Following chemical reconstitution of the [4Fe-4S], the N-His$_6$-SUMO-tag was cleaved by incubating the fusion protein with SenP1 protease (prepared in-house, 1 mg SenP1:100 mg Cfr) for 1 h at 10 °C in buffer containing 50 mM EPPS (pH 8.5), 300 mM KCl 15 % glycerol, and 5 mM DTT. The cleaved protein was purified away from SenP1 and the N-His$_6$-SUMO-tag by FPLC on a Mono Q 10/100 GL anion exchange column (GE Healthcare Life Sciences) using buffers containing 50 mM EPPS (pH 8.5), 50 mM or 1 M KCl (low-salt or high-salt), 15% glycerol, and 5 mM DTT. Protein was eluted using a linear gradient of 100% low-salt to 100% high-salt buffer over eight column volumes. Fractions containing apo-reconstituted, tag-less Cfr were

combined, concentrated using a concentrator cell (Amicon Ultra- 0.5 ml, 30 MWCO), and stored at –80°C. Protein concentration was determined by Bradford assay (Bio-Rad).

## Preparation of rRNA substrate

The *E. coli* 23S rRNA fragment 2447–2624 used for in vitro experiments was prepared using modified published protocols (*Stojković and Fujimori, 2015*). The desired DNA fragment was amplified from plasmid pKK3535 using previously established primers (*Yan et al., 2010*) and used as the template in the in vitro transcription reaction. Following DNase treatment and purification, RNA was precipitated overnight at –20°C by addition of 1/10th volume of 3 M NaOAc, pH 5.5, and 3 volumes of ethanol (EtOH). The RNA was then pelleted and washed with 70% aqueous EtOH, dried, and resuspended in nuclease-free water to obtain a final concentration of ~6 mg/ml. The rRNA fragment was refolded and purified by size exclusion chromatography. To refold the RNA, the sample was heated at 95°C for 2 min and then cooled to 65°C over 5 min. $MgCl_2$ was subsequently added to a final concentration of 10 mM prior to a final cooling step at room temperature for at least 30 min. After removing insoluble debris, RNA was purified by FPLC on a HiLoad 26/60 Superdex 200 column (GE Healthcare Life Sciences) using buffer containing 50 mM HEPES (pH 7.5), 10 mM $MgCl_2$, and 50 mM KCl. Fractions containing the desired rRNA product were combined and precipitated overnight at –20°C by addition of 1/10th volume of 3 M NaOAc, pH 5.5, and 3 volumes of EtOH. The RNA was then pelleted and washed with ice-cold 80% aqueous EtOH, dried, and resuspended in nuclease-free water. After confirming RNA purity on a denaturing 5% TBE, 7 M Urea-PAGE gel, the RNA sample was concentrated to ~450 mM using a SpeedVac Vacuum Concentrator prior to storage at –80°C.

## Cfr kinetic assay

Methylation activity of CfrWT and CfrV4 were assessed by monitoring radioactivity incorporation into RNA. Flavodoxin and flavodoxin reductase enzymes were prepared as described previously (*McCusker et al., 2012*). Prior to assembling reaction components, the RNA substrate was refolded as described above. Reactions were conducted in 52 μl volumes in an anaerobic chamber (MBraun, oxygen levels less than 1.8 ppm) under the following conditions: 100 mM HEPES (pH 8.0), 100 mM KCl, 10 mM $MgCl_2$, 2 mM DTT, 50 μM Flavodoxin, 25 μM Flavodoxin reductase, 100 μM rRNA substrate, 2 mM [$^3$H-methyl] *S*-adenosylmethionine (175.8 dpm/pmol), and 5 μM apo-reconstituted Cfr. Reactions were equilibrated at 37°C for 5 min and subsequently initiated by addition of NADPH (Sigma-Aldrich, final concentration 2 mM). The reaction was allowed to proceed at 37°C and timepoints at 0, 2, 4, 6, and 8 min of 10 μl volume were quenched by the addition of $H_2SO_4$ (50 mM final concentration). For each time point, the RNA volume was brought up to 100 μl with nuclease-free water and was purified away from other reaction components by an RNA Clean & Concentrator kit (Zymo Research) by following the manufacturer's instructions. Purified RNA eluate was added to Ultima Gold scintillation fluid, and the total amount of radioactivity incorporated in the product was detected using a Beckman–Coulter LS6500 scintillation counter. Amount of product generated at each time point was calculated by subtracting background radioactivity (t=0 min) and taking into account that two of the three tritium atoms from [$^3$H-methyl] *S*-adenosylmethionine would be incorporated into the final methylated RNA product (*Yan and Fujimori, 2011*; *Bauerle et al., 2018*).

## Evaluation of Cfr protein expression by quantitative western blot

*E. coli* expressing empty plasmid, CfrWT, or Cfr mutants were grown at 37°C to an $OD_{600}$ of ~0.4 with shaking by diluting an overnight culture 1:100 into 10 ml LB media containing ampicillin (100 μg/ml) and AHT inducer (30 ng/ml). Cells were harvested by centrifugation. Cell pellets were lysed for 15 min using B-PER Bacterial Protein Extraction Reagent (Thermo Fisher Scientific) containing DNase I (New England Biolabs) and 1× cOmplete, EDTA-free protease inhibitor cocktail (Roche). Whole-cell lysate samples containing 4 μg of protein were fractionated using a 4–20% SDS-PAGE gel (Bio-Rad). Proteins were transferred to a 0.2-μm nitrocellulose membrane using a Trans-Blot Turbo transfer system (Bio-Rad) with a 7 min, mixed MW protocol. Membranes were incubated with TBST-Blotto buffer (50 mM Tris-pH 7.5, 150 mM NaCl, 0.1% Tween-20, 5% w/v Bio-Rad Blotting Grade Blocker) for 1 hr at room temperature, followed by TBST-Blotto containing two primary antibodies: monoclonal mouse anti-FLAG M2 (1:2000 dilution, Sigma-Aldrich) and monoclonal rabbit anti-RNA polymerase beta (1:2000 dilution, Abcam) for 1 hr at room temperature. After washing 3× for 5 min with TBST, membranes

were then incubated overnight at 4°C with TBST-Blotto containing two secondary antibodies: goat anti-rabbit IgG cross-absorbed DyLight 680 (1:10,000 dilution, Thermo Fisher Scientific) and goat anti-mouse IgG cross-absorbed IRDye 800CW (1:10,000 dilution, Abcam). Membranes were rinsed 3× for 5 min with TBST and allowed to dry completely prior imaging using a Bio-Rad ChemiDoc Molecular Imager. Quantification was performed using Image Lab Software (Bio-Rad) within the linear range of detection. The housekeeping protein RNA polymerase beta, which was stably expressed in all experimental conditions, was used as an internal loading control.

## Determination of *E. coli* growth rate

*E. coli* expressing empty plasmid, CfrWT, or Cfr variants were grown at 37°C with shaking by diluting a 50 µl of an overnight culture into 10 ml of LB media containing ampicillin (100 µg/ml) and AHT inducer (30 ng/ml). $OD_{600}$ values were recorded every 20 min with a microtiter plate reader (SpectraMax M5, Molecular Devices).

## qPCR Primer Design and Validation

qPCR primer sequences for *cfr, recA,* and *luc* were designed using NCBI Primer Blast. Template accession numbers, amplicon length, and primer sequences are described in *Supplementary file 1*. Primer sequences for *rrsA* were used as published previously (*Zhou et al., 2011*). For each primer pair primer, qPCR was performed on a tenfold dilution series of desired samples. Amplification efficiency was calculated from the slope of the graph of Cq values plotted against $\log_{10}$ of the at least four template concentrations. Primers for recA: Y=–3.238*X+38.46, $R^2$=0.9992, PCR efficiency=103.6%. Primers for luc: Y=–3.316*X+34.52, $R^2$=0.9967, PCR efficiency=100.2%. Primers for cfr: Y=–3.254*X+37.52, $R^2$=0.9960, PCR efficiency=102.9%. Primers for rrsA: Y=–3.629*X+32.24, $R^2$=0.9965, PCR efficiency=90.0%.

## Determination of Cfr mRNA expression by RT-qPCR

### Bacterial growth

*E. coli* expressing empty plasmid control, CfrWT, or Cfr variants were grown at 37°C with shaking by diluting an overnight culture 1:100 into 5 ml of LB media containing ampicillin (100 µg/ml) and AHT inducer (30 ng/ml). When cells reached an $OD_{600}$ of ~0.4, RNAprotect Bacteria Reagent (QIAGEN) was added to the culture following the manufacturer's instructions. Cells were then harvested by centrifugation for 10 min at 5000×*g* at 4°C and frozen on dry ice.

### Total RNA isolation and DNase treatment

Pellets were then thawed and resuspended in 200 µl of lysis buffer containing 30 mM Tris-HCl (pH 8.0), 0.1 mM EDTA, 15 mg/ml lysozyme, and Proteinase K (New England Biolabs). Following lysis for 10 min at room temperature, total RNA was isolated using an RNeasy Mini Kit (QIAGEN) following the manufacturer's instructions. Yield and purity of isolated RNA were assessed by NanoDrop UV spectrophotometer (Thermo Fisher Scientific). RNA integrity was assessed by performing 1% TBE agarose gel electrophoresis with samples that had been boiled for 95°C for 5 min in RNA loading dye (New England Biolabs). Genomic DNA was eliminated by incubating 2 µg of RNA with 2 U of RQ1 RNase-free DNase I (Promega) for 30 min at 30°C. The DNase reaction was halted by the addition of RQ1 Stop Solution (Promega) and incubation for 10 min at 65°C.

### cDNA synthesis

Reverse transcription was performed using the iScript cDNA Synthesis Kit (Bio-Rad) following the manufacturer's instructions with tenfold diluted DNase-treated RNA. In brief, reactions of 20 µl volume were prepared by combining 4 µl 5× iScript buffer, 1 µl iScript RNase H + MMLV reverse transcriptase, 11 µl nuclease-free water, and 4 µl of RNA. Reactions were incubated for 5 min at 25°C, followed by 20 min at 42°C and 1 min at 95°C. If not used immediately, cDNA was stored at –20°C.

### RT-qPCR

SsoAdvanced Universal SYBR Green Supermix (Bio-Rad) was used for 10 µl qPCR reactions. Each reaction contained 5 µl of 2× Supermix, 0.3 µM of each forward and reverse primer, and 4 µl of diluted

cDNA. The cDNA was diluted 40-fold for reactions with *cfr* primers and 4000-fold for reactions *rrsA* primers. Reactions were prepared in a 96-well PCR Plate (Bio-Rad, MLL9601) and run on a Bio-Rad CFX qPCR Machine. The thermal cycling conditions were as follows: 98°C for 30 s, followed by 35 cycles of 98°C for 10 s and 60°C for 45 s with plate read, ending with melt curve analysis using 5 s, 0.5°C increment steps from 65°C to 95°C. A no template control and no reverse transcription control were included on each plate for each primer pair. Cq values were determined using CFX Maestro Software using a single threshold method. For each sample, the average of three triplicate Cq values was used for further analysis. Relative transcript expression was calculated using the Pfaffl method (*Pfaffl, 2001*). Expression was normalized to *rrsA* transcripts which is stably expressed in *E. coli* BW25113 (*Zhou et al., 2011*) and across our experimental conditions.

## Polysome analysis

Lysate preparation and sucrose gradient fractionation were adapted from previously published protocols with modification (*Mohammad and Buskirk, 2019*; *Li et al., 2014*).

### Lysate preparation

*E. coli* expressing empty plasmid control, CfrWT, or Cfr mutants were grown at 37°C with shaking by diluting an overnight culture 1:100 into 400 ml of LB media containing ampicillin (100 µg/ml) and AHT inducer (30 ng/ml). Cells were harvested at an $OD_{600}$ ~0.4–0.5 in 200 ml batches by rapid filtration at 37°C followed by flash freezing in liquid nitrogen as described previously (*Li et al., 2014*). Each frozen cell pellet was combined with 650 µl lysis buffer as frozen droplets containing 20 mM Tris (pH 8.0), 10 mM $MgCl_2$, 100 mM $NH_4Cl$, 5 mM $CaCl_2$, 0.4% Triton X-100, 0.1% NP-40, 100 U/ml RNase-free DNase I (Roche), and 10 U/ml SUPERase-In (Invitrogen). Cells with lysis buffer were pulverized in a 10-ml jar containing a 12-mm grinding ball using a TissueLyser II (QIAGEN) by performing five rounds of 3 min at 15 Hz. Canisters were pre-chilled by submersion in liquid nitrogen for at least 1 min prior to each round of pulverization. Lysates were recovered from the frozen jars using a spatula pre-chilled in liquid nitrogen and stored at –80°C until further use.

### Sucrose gradient fractionation

Pulverized lysates were thawed at 30°C for 2 min followed by an ice-water bath for 20 min. Lysates were clarified by centrifugation at 20,000×*g* for 10 min at 4°C. The RNA concentration of the clarified lysate was measured by NanoDrop UV spectrophotometer (Thermo Fisher Scientific) and diluted to 2.5 mg/ml with lysis buffer. Ribosome and mRNA components were separated on a linear, 12 ml, 10–55% (w/v) sucrose gradient containing 20 mM Tris (pH 8.0), 10 mM $MgCl_2$, 100 mM $NH_4Cl$, 2 mM DTT, and 10 U/ml SUPERase-In. Sucrose gradients were generated using a Bio-Comp Gradient Master with the following program: Time=1:58 s; Angle=81.5°, Speed=16 rpm. For each biological sample, 190 µl (~0.5 mg RNA) of clarified lysate was loaded onto sucrose gradients in duplicate. Ultracentrifugation was performed using a SW Ti41 rotor (Beckman Coulter) for 201,000×*g* for 2.5 hr at 4°C. Gradients were fractionated using a Bio-Comp Fractionator in 20 fractions at a speed of 0.25 mm/s where absorbance at 260 nm was continuously monitored.

### RNA extraction and DNase treatment

Fractions 1+ 2, 3+4, 16+17, and 18+19 were combined. RNA was extracted from uniform volumes of each fraction or combination of fractions. RNA extraction was performed by adding one volume of TRIzol reagent (Invitrogen), mixing until homogeneous, and incubating at room temperature for 5 min. Samples were then incubated at room temperature for another 5 min following the addition of 0.4 volumes of chloroform. After centrifugation for 15 min at 12,000×*g* at 4°C, the aqueous supernatant was transferred to a new tube to which 250 pg of a luciferase control RNA spike-in (luc, Promega). RNA was precipitated overnight at –20°C by the addition of 1 volume of isopropanol and 2 µl of GlycoBlue (15 mg/ml, Invitrogen). RNA was pelleted by centrifugation, washed two times with 75% ice-cold, aqueous ethanol, and allowed to dry at room temperature for ~30 min. The RNA was then resuspended in 20 µl of nuclease-free water. RNA quality and concentration were assessed by a NanoDrop UV spectrophotometer (Thermo Fisher Scientific). Genomic DNA was eliminated by incubating 10 µl of isolated RNA with 1 U of RQ1 RNase-free DNase I (Promega) for 30 min at 30°C.

The DNase reaction was halted by the addition of RQ1 Stop Solution (Promega) and incubation for 10 min at 65°C.

## cDNA synthesis and RT-qPCR

Reverse transcription was performed using the iScript cDNA Synthesis Kit (Bio-Rad) following the manufacturer's instructions. In brief, reactions of 20 µl volume were prepared by combining 4 µl 5× iScript buffer, 1 µl iScript RNase H + MMLV reverse transcriptase, 5 µl nuclease-free water, and 10 µl of DNase-treated RNA. Reactions were incubated for 5 min at 25°C, followed by 20 min at 42°C and 1 min at 95°C. SsoAdvanced Universal SYBR Green Supermix (Bio-Rad) was used for 10 µl qPCR reactions in a 96-well plate as described above. Each reaction contained 5 µl of 2× Supermix, 0.3 µM of each forward and reverse primer, and 4 µl of tenfold diluted cDNA. Reactions containing cfr, recA, and luc primers (*Supplementary file 1*) were performed for each fraction, including a no template control and no reverse transcription control for each primer set on each plate. The average of three triplicate Cq values was used for further analysis.

## Data analysis

Normalized mRNA distribution profiles for the target mRNAs were calculated as described previously (*Pringle et al., 2019*). In brief, the relative abundance of each target mRNA normalized to luciferase RNA spike-in. The percentage of target mRNA found across gradient fractions was calculated by dividing the amount of target mRNA detected in one fraction by the sum of the target mRNA detected in all fractions.

## Protein degradation assay

### Bacterial growth and rifampicin treatment

*E. coli* expressing CfrWT or Cfr mutants were grown at 37°C with shaking by diluting an overnight culture 1:100 into 25 ml of LB media containing ampicillin (100 µg/ml) an AHT inducer (30 ng/ml). When cells reached an $OD_{600}$ ~0.4–0.5, rifampicin (Sigma-Aldrich) was subsequently added to a final concentration of 100 µg/ml, and cultures were allowed continued incubation at 37°C with shaking. Time points at 0, 20, 40, 60, 80, and 100 min were harvested by centrifuging 3 ml of the culture at 8000 rpm at 4°C for 10 min, decanting the supernatant, and immediately flash-freezing the pellet in liquid nitrogen. Cell pellets for each time point were lysed using B-PER Bacterial Protein Extraction Reagent as described above.

### Western blot

Whole-cell lysate samples containing 5 µg of protein were fractionated on a 4–20% SDS-PAGE gel and transferred onto a 0.2-µm nitrocellulose membrane as described above. Membranes were stained with Ponceau S stain (0.1% w/v Ponceau S, 5% v/v acetic acid) and imaged using a Bio-Rad ChemiDoc Molecular Imager. After blocking in TBST-Blotto buffer for 1 hr at room temperature, membranes were incubated with TBST-Blotto containing primary monoclonal mouse anti-FLAG M2 antibody (1:2000 dilution, Sigma-Aldrich) or monoclonal mouse anti-GAPDH antibody (1:2000 dilution, Abcam) for 1 hr at room temperature. After washing 3× for 5 min with TBST, membranes were incubated overnight at 4°C with TBST-Blotto containing a secondary antibody, goat anti-mouse cross-absorbed IRDye 800CW (1:10,000 dilution, Abcam). Membranes were rinsed and imaged as described above.

### Data analysis

Quantification was performed using Image Lab Software (Bio-Rad) within the linear range of detection. The Ponceau S total protein stain was used to normalize for differential sample loading. Percentage (%) of initial Cfr protein remaining was calculated by dividing the amount of Cfr protein at a given time point after rifampicin treatment by the amount of Cfr protein prior to rifampicin treatment (t=0 min) .

## Purification of Cfr-modified *E. coli* ribosome

Cfr-modified, 70S ribosomal subunit was purified from *E. coli* MRE600 expressing CfrV7 variant using previously published protocol with modification (*Mehta et al., 2012*; *Stojković et al., 2020*). In short, *E. coli* transformed with pZA-encoded CfrV7 were grown to an $OD_{600}$ of 0.5 in LB media containing

ampicillin (100 μg/ml) and AHT inducer (30 ng/ml) at 37°C with shaking. Cells were harvested by centrifugation, washed, and lysed by using a microfluidizer. The lysate was clarified by ultracentrifugation at 30,000×*g* 30 min at 4°C using a Ti45 rotor (Beckman Coulter) two times. The recovered supernatant was applied to a 32% w/v sucrose cushion in buffer containing 20 mM Hepes-KOH (pH 7.5), 500 mM $NH_4Cl$, 20 mM $Mg(OAc)_2$, 0.5 mM EDTA, 6 mM β-mercaptoethanol, 10 U/ml SuperASE-In and was ultracentrifuged at 100,000×*g* for for 16 hr at 4°C in a SW Ti41 rotor (Beckman Coulter). After removing the supernatant, the pellet was resuspended slowly at 4°C over 1 hr in Buffer A containing 20 mM Hepes-KOH (pH 7.5), 200 mM $NH_4Cl$, 20 mM $Mg(OAc)_2$, 0.1 mM EDTA, 6 mM β-mercaptoethanol, and 10 U/ml SuperASE-In. Particulates that were not resuspended were removed by centrifugation at 10,000 rpm for 10 min at 4°C. Sample concentration was determined by NanoDrop UV spectrophotometer (Thermo Fisher Scientific), where $A_{260}$=1 corresponds to 24 pmol of 70S ribosome. Tight-coupled 70S ribosomes were purified as described previously (*Khusainov et al., 2017*). In brief, 70S ribosomes were purified on a 15–30% w/v sucrose gradient in Buffer A. Sucrose gradients were generated using a Bio-Comp Gradient Master. 300–400 pmol of 70S ribosomes were loaded on each sucrose gradient. Ultracentrifugation was performed using a SW Ti41 rotor (Beckman Coulter) for 75,416×*g* for 16 hr at 4°C. Gradients were fractionated using a Bio-Comp Fractionator in 20 fractions at a speed of 0.25 mm/s where absorbance at 260 nm was continuously monitored. Fractions corresponding to 70S ribosomes were combined and precipitated by slow addition at 4°C of PEG 20,000 in Buffer A to a final concentration of 9% w/v. Ribosomes were isolated by centrifugation for 10 min at 17,500×*g*. After removing the supernatant, ribosomes were slowly resuspended overnight at 4°C in buffer containing 50 mM Hepes-KOH (pH 7.5), 150 mM KOAc, 20 mM $Mg(OAc)_2$, 7 mM β-mercaptoethanol, 20 U/ml SuperASE-In.

## Cryo-EM analysis

Purified 70S ribosomal subunits were diluted from 2 to 0.5 mg/ml in Buffer A, applied to 300-mesh carbon coated (2 nm thickness) holey carbon Quantifoil 2/2 grids (Quantifoil Micro Tools) and flash-frozen as described in *Khatter et al., 2015*. Data were collected using serialEM on the in-house Titan Krios X-FEG instrument (Thermo Fisher Scientific) operating at an acceleration voltage of 300 kV and a nominal underfocus of Δz=0.2–1.5 μm at a nominal magnification of 29,000 (calibrated physical pixel size of 0.822 Å). We recorded 2055 movies using a K2 direct electron detector camera in super-resolution mode with dose fractionation (80 individual frames were collected, starting from the first one). Total exposure time was 8 s, with a total dose of 80 e- (or 1 e-/Å2/frame). Images in the stack were aligned using the whole-image motion correction and patch motion correction (5×5 patches) methods in MotionCor2 (*Zheng et al., 2017*). Before image processing, all micrographs were checked for quality and 1531 best were selected for the next step of image processing. The contrast transfer function of each image was determined using GCTF (*Zhang, 2016*) as a standalone program. For particle selection, we have used Relion 3.0 autopicking procedure (*Scheres, 2012*). For the first steps of image processing, we used data binned by a factor of 8 (C8 images). During the first round of 2D classification, we removed only images with ice or other contaminants. Subsequently, the initial structure was generated using the ab initio procedure in CryoSPARC v2.0. Following this step, we performed Relion 3D classification with bin by four data (C4) in order to exclude bad particles. The resulting 141,549 particle images of ribosomes were used for subsequent classification and refinement procedures. For the initial refinement, we used a spherical mask, which was followed by further refinement using a mask around the stable part of 50S (excluding L1 stalk, L7/L12 region). A further improved cryo-EM map was obtained by using CTF-refinement procedure from Relion 3.0. The post-processing procedure implemented in Relion 3.0 (*Scheres, 2012*) was applied to the final maps with appropriate masking, B-factor sharpening (automatic B-factor estimation was –55.86) and resolution estimation to avoid over-fitting (final resolution after post-processing with 50S mask applied was 2.7 Å). Subsequently, the stack of CTF-refined particles was processed in a new version of CryoSPARC v2.0 (*Punjani et al., 2017*). After homogeneous refinement, the same stack of particles was additionally refined in cisTEM (*Grant et al., 2018*). After Auto-Refine (with automasking within cisTEM), we performed local refinement using 50S mask (the same one used for refinement in Relion) and also applied per particle CTF refinement as implemented in cisTEM software. After such refinement, the resolution was improved to 2.2 Å (*Figure 5—figure supplement 1*). This map after Sharpen3D (*Grant et al., 2018*) was used for model building and map interpretation.

## Atomic model building and refinement

The final model of the 50S subunit was generated by multiple rounds of model building in Coot (*Emsley et al., 2010*) and subsequent refinement in PHENIX (*Adams et al., 2010*). The restraints for the novel $m^2m^8A$ nucleotide for the atomic model fitting and refinements were generated using eLBOW (*Moriarty et al., 2009*). The atomic model of the 50S subunit from the *E. coli* ribosome structure (PDB 6PJ6) (*Stojković et al., 2020*) was used as a starting point and refined against the experimental cryo-EM map by iterative manual model building and restrained parameter-refinement protocol (real-space refinement, positional refinement, and simulated annealing). Final atomic model comprised of ~92,736 atoms (excluding hydrogens) across the 3015 nucleotides and 3222 amino acids of 28 ribosomal proteins. Proteins L7, L10, L11, and L31 were not modeled in. In addition, 179 $Mg^{2+}$, 2716 water molecules, 1 $Zn^{2+}$, and 1 $Na^+$ were included in the final model. Prior to running MolProbity (*Chen et al., 2010*) analysis, nucleotides 878–898, 1052–1110, 2101–2189 of 23S rRNA, and ribosomal protein L9 were removed, due to their high degree of disorder. Overall, protein residues and nucleotides show well-refined geometrical parameters (*Table 1*). Figures were prepared using Pymol Molecular Graphics System, Version 2.4.1 unless otherwise noted.

## qPTxM analysis of post-transcriptional modifications

The final model and map were run through qPTxM (*Stojković et al., 2020*) with default parameters except for d_min=2 and cc_threshold=0.5 to search for evidence of posttranscriptional modifications. Of a total of 39 sites with density suggesting possible modifications, 2 were C8-methyl adenosines, A556 and A2503. None of the identified sites were 2'O-methyl cytosines. To calculate expected density dropoff curves for methylated and unmethylated nucleotides, the *phenix.fmodel* (*Adams et al., 2010*) tool was used to generate noise-free maps from models of a single nucleotide in each state, and scripts modified from qPTxM were used to collect measurements of the density at 0.1 Å intervals along the vector of the proposed methylation. Means and standard deviations were calculated for densities at the four positions tested by qPTxM on each nucleotide, from which Z-scores were then calculated for selected nucleotides. To measure densities for both the best tested rotamer of m(2'O)C 2498 and the modeled rotamer, densities along the 2'O-methyl bond were compared between the files generated by qPTxM run two times as described above, once with prune=True (removing the modeled methyl group and placing the rotameric methyl with the strongest density) and once with prune=False (leaving the modeled methyl group intact).

## Acknowledgements

The authors thank members of the Fujimori lab for discussion and comments on the manuscript. The authors thank the UCSF Center for Advanced CryoEM, which is supported by the National Institutes of Health (S10OD020054 and 1S10OD021741) and the Howard Hughes Medical Institute (HHMI). The authors thank Professor Kim Lewis and Dr. Yu Imai of Northeastern University for kindly providing hygromycin A.

## Additional information

### Competing interests

Adam Frost: Reviewing editor, *eLife*. The other authors declare that no competing interests exist.

### Funding

| Funder | Grant reference number | Author |
| --- | --- | --- |
| National Institute of Allergy and Infectious Diseases | R01AI137270 | Danica Galonić Fujimori |
| National Science Foundation | 1650113 | Kaitlyn Tsai |
| University of California, San Francisco | | Kaitlyn Tsai |

| Funder | Grant reference number | Author |
| --- | --- | --- |
| National Institute of General Medical Sciences | F32GM133129 | Iris D Young |
| National Institute of General Medical Sciences | R01GM123159 | James S Fraser |
| Sanghvi-Agarwal Innovation Award | | James S Fraser |

The funders had no role in study design, data collection and interpretation, or the decision to submit the work for publication.

## Author contributions

Kaitlyn Tsai, Conceptualization, Data curation, Formal analysis, Funding acquisition, Investigation, Methodology, Visualization, Writing – original draft, Writing – review and editing; Vanja Stojković, Conceptualization, Data curation, Formal analysis, Investigation, Methodology, Writing – review and editing; Lianet Noda-Garcia, Alexander G Myasnikov, Data curation, Formal analysis, Methodology, Writing – review and editing; Iris D Young, Data curation, Formal analysis, Writing – original draft, Writing – review and editing; Jordan Kleinman, Data curation, Writing – review and editing; Ali Palla, Data curation, Methodology, Writing – review and editing; Stephen N Floor, Formal analysis, Methodology, Writing – review and editing; Adam Frost, Formal analysis, Methodology, Resources, Supervision, Writing – review and editing; James S Fraser, Data curation, Formal analysis, Resources, Supervision, Writing – review and editing; Dan S Tawfik, Formal analysis, Funding acquisition, Methodology, Resources, Supervision, Writing – review and editing; Danica Galonić Fujimori, Conceptualization, Formal analysis, Funding acquisition, Project administration, Resources, Supervision, Writing – review and editing

## Author ORCIDs

Kaitlyn Tsai ⓘ http://orcid.org/0000-0002-0605-2720
Vanja Stojković ⓘ http://orcid.org/0000-0002-9885-3428
Iris D Young ⓘ http://orcid.org/0000-0003-4713-9504
Jordan Kleinman ⓘ http://orcid.org/0000-0001-7146-824X
Ali Palla ⓘ http://orcid.org/0000-0001-7380-3881
Stephen N Floor ⓘ http://orcid.org/0000-0002-9965-9694
Adam Frost ⓘ http://orcid.org/0000-0003-2231-2577
James S Fraser ⓘ http://orcid.org/0000-0002-5080-2859
Dan S Tawfik ⓘ http://orcid.org/0000-0002-5914-8240
Danica Galonić Fujimori ⓘ http://orcid.org/0000-0002-4066-9417

## Decision letter and Author response

Decision letter https://doi.org/10.7554/eLife.70017.sa1
Author response https://doi.org/10.7554/eLife.70017.sa2

## Additional files

### Supplementary files

• Transparent reporting form
• Supplementary file 1. Primer sequences, directed evolution mutants, and MIC values for FLAG constructs.

### Data availability

Atomic coordinates have been deposited in the Protein Data Bank under accession number 7LVK, and the density map has been deposited in the EMDB under accession number 23539.

The following dataset was generated:

| Author(s) | Year | Dataset title | Dataset URL | Database and Identifier |
|---|---|---|---|---|
| Stojkovic V, Myasnikov AG, Frost A, Fujimori DG | 2021 | Cfr-modified 50S subunit from Escherichia coli | https://www.rcsb.org/structure//7LVK | RCSB Protein Data Bank, 7LVK |
| Stojkovic V, Myasnikov AG, Frost A, Fujimori DG | 2021 | Cfr-modified 50S subunit from Escherichia coli | https://www.ebi.ac.uk/emdb/EMD-23539 | Electron Microscopy Data Bank, EMD-23539 |

The following previously published datasets were used:

| Author(s) | Year | Dataset title | Dataset URL | Database and Identifier |
|---|---|---|---|---|
| Stojkovic V, Myasnikov A, Frost A, Fujimori DG | 2020 | High resolution cryo-EM structure of E.coli 50S | https://www.rcsb.org/structure/6PJ6 | RCSB Protein Data Bank, 6PJ6 |

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
