## [Editor Report]

The paper addresses an important unresolved mechanism of antibiotic resistance caused by a RNA-modifying enzyme Cfr, a protein that confers resistance to multiple ribosome-targeting antibiotics due to methylation of the rRNA residue A2503 on the large ribosomal subunit. Tsai et al. identify the mutational hotspots that increase Cfr activity and show how a single methyl group on A2503 can hamper the antibiotic binding. The paper is an important contribution to our understanding of antibiotic resistance and is of broad interest to readers in the field of antibiotic resistance, biochemistry, structural biology, and medicinal chemistry.

---

## [Decision Letter]

Thank you for submitting your article "Directed evolution of the rRNA methylating enzyme Cfr reveals molecular basis of antibiotic resistance" for consideration by *eLife*. Your article has been reviewed by 3 peer reviewers, one of whom is a member of our Board of Reviewing Editors, and the evaluation has been overseen by a Reviewing Editor and Michael Marletta as the Senior Editor. The reviewers have opted to remain anonymous.

Essential revisions:

The referees agreed that the experiments were carefully performed and interpreted. The primary experimental revision (point 1) relates to providing a suitable control that tRNA abundance determines the higher translational efficiency of the mutant Cfr constructs. There are also suggestions for further experiments that would strengthen the paper (points 2, 4, 5), but we feel this is optional and could be viewed as beyond the scope of this particular study. I therefore leave it to you whether you perform these additional experiments (which might well be interesting).

1. Directed evolution yielded protein isoforms that are better expressed and are more stable. The authors show that mutant mRNAs are shifted towards heavier polysomes and suggest that this may be due to the tRNA abundance effect (Figure 4 – supplement 1). However, this experiment lacks a control. If tRNA abundance is crucial, overexpression of the tRNA decoding the wt AAU codon should also increase protein expression.

2. The authors find two shortened Cfr isoforms that apparently arise from incorrect initiation on internal methionine codons on cfr mRNA (I26M, M95) and show that translation can start at an initiation codon upstream of M1. This is by itself an exciting story in the Cfr story. The regulatory mechanism due to an uORF is plausible and well documented in Figure 4- supplement 3 and 4. However, the appearance of a (native) internal initiation site starting from M95 is surprising. Given that this is a very prominent translation product (well over 50% with the native Pcfr, Figure 4 – supplement 4), this aspect of the paper raises questions. How does the initiation from M95 depend on antibiotics? The authors clearly show that the shorter isoform does not confer resistance and is rapidly degraded. So, what is the functional role of the shorter isoform? To address these questions the authors could control how much Cfr-short isoform (M95) is present in the absence of antibiotics and how it is affected upon antibiotic treatment. Figures 4-supplementary 3 and 4 are so interesting that I would suggest including them in the main text. The expression of Cfr short (M95) seems to depend on the N2 mutation. This should be quantified and explained.

3. There is a rich literature (cited on p. 21 of the discussion) that may suggest alternative explanations. The authors should identify which of these suggestions can be excluded based on their data and clearly describe those that could contribute to the observed expression effect.

The text on p. 21, lines 6-19 should be modified, as it does not exactly state what was actually suggested, there are too many citations without clearly stating what these papers say.

4. The increased stability of the mutated Cfr variants is fascinating, but not well addressed. What causes the increase in protein stability? What is actually the N-terminus of the protein: is it deformylated and the N-terminal Met cleaved as efficiently as the wt? Is there any correlation with the N-end rule? One way to answer these questions would be to check the N-terminus of proteins from the gels by mass spectrometry. Add a few sentences in the Discussion as to how a (presumably N-terminal) Lys or Ile can increase protein stability.

5. Cfr methylates C8 of A2503. Mass spectra clearly distinguish the peaks of m2m8A2503 and m2A2503. However, this appears to rely on the assumption that A2503 is completely m2-modified. Has this been checked? What would be the mass of the unmodified fragment and was it identified in the spectra? Can the authors exclude the appearance of m8A2503 species? Please comment on the modification completeness of m2A2503 and provide the expected m/z value for the respective unmodified RNA fragment.

6. Since the ribosome structure presented is of a modified *E. coli* ribosome and the cfr enzyme is originally from S. aureus, the sequence alignment of Cfr from S. aureus and *E. coli* should be presented (as a Supplement). The authors should discuss whether it is legitimate to use S. aureus cfr to modify *E. coli* ribosome. This would widen the scope of the paper by generalizing the conclusions for gram-positive and gram-negative bacteria. This could be addressed in the introduction (e.g., p. 4 line 9 and line 15) and in the Results section (e.g., p.16 line 4 and line 9). In Figure 5 – add to the legend that this is an *E. coli* ribosome. In addition, please relate to this point in the discussion (p. 20).

7. The authors superimposed the complex structure of an archeal ribosome and linezolid and tiamulin for explaining the resistance mechanism. However, this resistance mechanism was not found in archaea and thus these ribosomes are less relevant for clinical aspects. More appropriate models to be used are of bacterial ribosomes in complex with these antibiotics, i.e. D. radiodurans ribosome in complex with tiamulin (PDBID 1XBP) and the pathogen *S. aureus* ribosome in complex with linezolid (PDBID 4WFA).

8. The paper would benefit from an analysis of the natural variability of the evolved sites. It is mentioned that clinical isolates do not have these exact substitutions, but a supplemental multiple sequence alignment of homologues with these positions marked would anyway be interesting. For example, an N2K variant appears in the results of a quick blastp search in a Bacillus Cfr.

9. Another remaining question is how the novel variants affects susceptibilities to ribosome-targeting antibiotics in other structural classes, including nucleoside analogues and macrolides. In the introduction the authors mention that in addition to PhLOPSA antibiotics, Cfr can confer resistance to nucleoside analog A201A, hygromycin A, and 16-membered macrolides. However, this is not returned to in the paper. MIC experiments with an expanded selection of antibiotics would be really nice to see if possible. Similarly, does the structure explain the previously observed cross-resistance that goes beyond PhLOPS(A)?

*Reviewer #1:*

The paper by Tsai et al. describes the antibiotic resistance mechanism caused by an RNA-modifying enzyme Cfr. The paper is important, as Cfr confers resistance to eight classes of antibiotics (collectively known as PhLOPSA). The strength of the manuscript is in the combination of different experimental techniques. These include directed evolution of Cfr; mass spectrometry analysis of the modification site in the 23S rRNA; mutational studies with quantitative analysis of their effect on the mRNA and protein levels of Cfr in the cell and its link to resistance; and finally, a high-resolution of cryo-EM of the ribosome with the fully modified m2m8A2503, the product of Cfr activity. Using directed evolution, the authors identified two mutational hotspots that increased resistance, one due to a more active promotor and another due to a mutation of residue N2 in Cfr. They demonstrate that Cfr methylates C8 of m2A2503 and that Cfr mutants are more efficient in installing the m8 modification. However, the catalytic activity of the mutants is not affected. Rather, the cellular concentration of Cfr is increased due to higher translation efficiency and higher Cfr protein stability. Finally, the cryo-EM structure clearly shows the modified nucleotide, which provides explanations as to how a single methyl group causes resistance to so many antibiotics.

The work is elegantly designed, and the experiments are of high technical quality. The paper is very well written and most of the conclusions are based on solid data. However, there are some loose ends, which should be addressed either experimentally or at least in the discussion.

1. The authors find two shortened Cfr isoforms that apparently arise from incorrect initiation on internal methionine codons on cfr mRNA (I26M, M95) and show that translation can start at an initiation codon upstream of M1. This is by itself an exciting story in the Cfr story. The regulatory mechanism due to an uORF is plausible and well documented in Figure 4- supplement 3 and 4. However, the appearance of a (native) internal initiation site starting from M95 is surprising. Given that this is a very prominent translation product (well over 50% with the native Pcfr, Figure 4 – supplement 4), this aspect of the paper raises questions. How does the initiation from M95 depend on antibiotics? The authors clearly show that the shorter isoform does not confer resistance and is rapidly degraded. So, what is the functional role of the shorter isoform? These are very exciting questions that should be addressed.

2. Directed evolution yielded protein isoforms that are better expressed and are more stable. The authors show that mutant mRNAs are shifted towards heavier polysomes and suggest that this may be due to the tRNA abundance effect (Figure 4 – supplement 1). However, this experiment lacks a control. If tRNA abundance is crucial, overexpression of the tRNA decoding the wt AAU codon should also increase protein expression. There is a rich literature (cited on p. 21 of the discussion) that may suggest alternative explanations. The authors should identify which of these suggestions can be excluded based on their data and clearly describe those that could contribute to the observed expression effect.

3. The increased stability of the mutated Cfr variants is fascinating, but not well addressed. What causes the increase in protein stability? What is actually the N-terminus of the protein: is it deformylated and the N-terminal Met cleaved as efficiently as the wt? Is there any correlation with the N-end rule?

4. Cfr methylates C8 of A2503. Mass spectra clearly distinguish the peaks of m2m8A2503 and m2A2503. However, this appears to rely on the assumption that A2503 is completely m2-modified. Has this been checked? What would be the mass of the unmodified fragment and was it identified in the spectra? Can the authors exclude the appearance of m8A2503 species?

My suggestion to address points 1-4 is the following.

1. The authors could control how much Cfr-short isoform (M95) is present in the absence of antibiotics and how it is affected upon antibiotic treatment. Figures 4-supplementary 3 and 4 are so interesting that I would suggest including them in the main text. The expression of Cfr short (M95) seems to depend on the N2 mutation. This should be quantified and explained.

2. An experiment with overexpression of tRNA decoding the wt AAU codon would provide an important test for the authors' suggestion that the expression differences are related to tRNA abundance. The text on p. 21, lines 6-19 should be modified, as it does not exactly state what was actually suggested, there are too many citations without clearly stating what these papers say.

3. Check the N-terminus of proteins from the gels by mass spectrometry. Add a few sentences in the Discussion as to how a (presumably N-terminal) Lys or Ile can increase protein stability.

4. Comment on the modification completeness of m2A2503 and provide the expected m/z value for the respective unmodified RNA fragment.

*Reviewer #2:*

The manuscript deals with a critical resistance mechanism relevant to eight families of antibiotics that inhibit ribosome activity. In this mechanism, the enzyme Cfr methylates the rRNA nucleotide 2503, which is part of the typical binding pocket of these antibiotics. By modifying the binding pocket architecture, binding of the drug is inhibited, and resistance to bacterial antibiotics is achieved. The authors aimed to decipher the resistance mechanism by structural studies, but the moderate Cfr enzyme activity of *S. aureus* hampered the ability to study such methylated ribosomes.

The authors decipher how antibiotic resistance can be increased by introducing mutations into the modifier enzyme Cfr by using a directed evolution approach. Using this approach, a high percentage of mutated ribosomes suitable for structural study were achieved. They discover that the mutated Cfr enzyme is not more efficient in methylation, but rather that the mutations alter active intercellular protein levels, and therefore higher methylation ratio is achieved. In addition, they found that the introduced mutations in Cfr enhance its translation efficiency as it is shown to be part of well-translated mRNAs, and this is the leading cause for higher methylation levels of the mutated Cfr. For some of their Cfr mutants, it is demonstrated that the increase in the expression of mutated Cfr is due to an improved translation that is mediated by tRNA abundance. The Cfr mutants show increased half-life, due to enhanced translation and decreased degradation. This part of the paper is the major technical advance.

The cryo-EM 2.2 A structure of an *E. coli* ribosome with a C8-modified 2503 rRNA nucleotide was obtained by using the mutated stable Cfr, which enables ~90% of 2503 methylation. Furthermore, by superimposing few published available antibiotics-ribosome complex structures on the current new mutated ribosome structure, the authors provide a valid explanation of the resistance mechanism. They demonstrate how the addition of a methyl group at the C8 position of A2503 will cause a steric clash with a bound drug and preclude its binding. This single structural change in the ribosome upon methylation may indeed be used for drug design for overcoming antibiotic resistance.

1. It should be mentioned that the cfr mutations gained using directed evolution are located in sites of the proteins which are distant from its active site. The most effective mutation (in residue 2) is in a separate domain from the active site. Actually, all the mutations are mainly at the surface of the Cfr. This is clear from Figure 1b and should be added.

2. Since the ribosome structure presented is of a modified *E. coli* ribosome and the cfr enzyme is originally from S. aureus the sequence alignment of Cfr from S. aureus and *E. coli* should be presented (as a supp.) and discussed.

How similar they are? Is it legitimate to use S. aureus cfr to modify *E. coli* ribosome? Apparently, it is since the authors present evidence for it but some text to relate to this point should be added. It will also widen the scope by generalizing the conclusions for gram-positive and gram-negative bacteria.

In the text, this point should be addressed in the introduction (For example: P. 4 line 9. Cfr from MRSA and in line 15 *E. coli* ribosome.) and in the Results section (For example: p.16 line 4 – mention S. aureus ribosome but the same page line 9 – the ribosomes of which the structure was determined are EC ones. In Figure 5 – add to the legend that this is an EC ribosome. In addition, please relate to this point in the discussion (p.20)).

3. The authors superimposed the complex structure of an archeal ribosome and linezolid and tiamulin for explaining the resistant mechanism. Resistant mechanism was not found in archea and this ribosome is less relevant for clinical aspects. More appropriate models to be used are of a bacterial ribosome in complex with these antibiotics, i.e. D. radiodurans ribosome in complex with tiamulin (PDBID 1XBP) and the pathogen *S. aureus* ribosome in complex with linezolid (PDBID 4WFA).

*Reviewer #3:*

In this study, Tsai et al. set out out to generate Cfr variants through directed evolution that are more effective in rRNA methylation and therefore resistance. The authors achieved this aim and exploited the novel variants to generate modified ribosomes that were suitable substrates for cryo-EM structural analysis. The paper is clear, and the data well presented. Directed evolution of Cfr was clearly an excellent strategy for generating near-stoichiometrically modified ribosomes. The evolved sites of Cfr are not very interesting in respect to protein function per se, as their effect seems mostly about increasing expression levels rather than altering enzymatic activity. However, these variants were clearly invaluable for enabling the authors to fulfil their aim of generating a high-resolution structure. By comparing this structure to other ribosome structures that include bound antibiotics, the authors show clearly how the modification is incompatible with antibiotic binding.

Overall, the paper is a significant contribution to our understanding of the mechanistic basic of resistance to a PhLOPS(A) antibiotics as conferred by the clinically relevant resistance factor Cfr. I expect the structure will be used by many in the ribosome field.

---

## [Author Response]

Essential revisions:The referees agreed that the experiments were carefully performed and interpreted. The primary experimental revision (point 1) relates to providing a suitable control that tRNA abundance determines the higher translational efficiency of the mutant Cfr constructs. There are also suggestions for further experiments that would strengthen the paper (points 2, 4, 5), but we feel this is optional and could be viewed as beyond the scope of this particular study. I therefore leave it to you whether you perform these additional experiments (which might well be interesting).1. Directed evolution yielded protein isoforms that are better expressed and are more stable. The authors show that mutant mRNAs are shifted towards heavier polysomes and suggest that this may be due to the tRNA abundance effect (Figure 4 – supplement 1). However, this experiment lacks a control. If tRNA abundance is crucial, overexpression of the tRNA decoding the wt AAU codon should also increase protein expression.

The reviewer brings up an excellent point. To address it, we have carried out an in-depth investigation. To perform tRNA overexpression experiments, we cloned the tRNA^Asn^ gene, which decodes the WT AAU codon, into a pBSTNAV vector. The pBSTNAV vector is commonly used for overexpression of tRNAs in *E. coli* and is driven by a strong, constitutive *lpp* promoter (Dégut et al., 2016). To ensure that the pBSTNAV_tRNA^Asn^ vector was suitable for coexpression with the pZA-encoded Cfr vectors, we swapped the pBSTNAV ampicillin resistance marker for kanamycin. The pBSTNAV and pZA vectors are also ori-compatible. The sequence architecture of the construct is outlined in Author response image 1:

**Author response image 1. sa2fig1:** pBSTNAV_tRNA^Asn^ construct.

To perform co-expression experiments, *E. coli* BW25113 was co-transformed with pZA (Empty, CfrWT-FLAG, or CfrN2K-FLAG) and pBSTNAV (Empty or tRNA^Asn^). We confirmed tRNA overexpression by northern blot (Author response image 2) . We designed the 5’-biotinylated DNA oligo probe (purchased from IDT) to be complementary to the region of tRNA^Asn^ containing the anticodon stem loop. The sequence is as follows: 5’-Biotin-GATTAACAGTCCGCCGTTCTACCGAC. Of note, we observe additional bands in the tRNA overexpression samples, likely attributed to pre-processed or truncated tRNA^Asn^.

**Author response image 2. sa2fig2:** Confirmation of tRNA^Asn^ overexpression. Trizol-extracted total RNA was extracted from bacterial lysates and separated on a denaturing 10% acrylamide 8M TBE-Urea gel. Em = empty vector and WT = CfrWT-FLAG. (a) RNA was stained with SYBR-safe. (b) For detection of tRNA^Asn^ by northern blot, RNA was transferred onto a Hybond N+ nylon membrane (GE Healthcare) and crosslinked to the membrane using a Stratalink 1800 UV Crosslinker. After blocking for 2.5 h at 40^o^C in hybridization buffer (250 mM sodium phosphate buffer-pH 7.0, 1 mM EDTA, 7% (w/v) SDS, 0.5% (w/v) BSA, 4 µg/mL salmon sperm DNA), the membrane was incubated with denatured tRNA probe overnight at 40^o^C in hybridization buffer. After washing the membrane 3 X 15 min in 1X SSC, 0.1% SDS, tRNA^Asn^ was detected using the Thermo Chemiluminescent Nucleic Acid Detection Module following manufacturer’s instructions.

To test how overexpression of tRNA^Asn^ impacts Cfr protein levels, co-transformants were grown to mid-log phase in LB media containing selection antibiotics and AHT inducer (30 ng/mL). Cfr protein levels were detected by immunoblotting against the FLAG tag as described in materials and methods. While performing the experiment, it became evident that co-expression of tRNA^Asn^ caused significant fitness defects. Cultures that co-expressed tRNA^Asn^ required 20 min longer (2h 5 min vs. 1h 45 min) to reach mid-log phase than those expressing an empty pBSTNAV vector. This result is also evident in the western outlined in Author response image 3, where overexpression of tRNA^Asn^ causes decreased expression of both CfrWT and CfrN2K (Author response image 3) .

**Author response image 3. sa2fig3:** Overexpression of tRNA^Asn^ decreases Cfr protein levels. Cfr protein was detected by immunoblotting against the FLAG tag and RNA polymerase β subunit was used as the loading control. Em = empty vector, WT = CfrWT-FLAG, and N2K = CfrN2K-FLAG.

This result suggests that overexpression of tRNA alone may not be sufficient for proper tRNA production and modulation of protein synthesis. Maturation of tRNA^Asn^ requires trimming of 3’ and 5’ pre-tRNA sequence by RNases, post-transcriptional modification by various enzymes, and charging by asparaginyl-tRNA synthetase (Shepherd and Ibba, 2015). Thus, overexpression tRNA^Asn^ may overwhelm endogenous tRNA maturation machinery, leading to accumulation of potentially toxic pre-processed intermediates and partially degraded tRNA^Asn^ (Author response image 2) , along with uncharged tRNA^Asn^. Additionally, if there did exist an increased level of charged Asn-tRNA^Asn^ in our overexpression system, previous work suggests that disrupting the balance of the charged tRNA pool competing for the ribosomal A-site negatively impacts the fidelity of protein synthesis (Kramer and Farabaugh, 2007). Together, these results indicate that while in theory overexpression of tRNA^Asn^ would allow additional testing of our model, the fitness defects associated with this experimental setup makes interpretation of results challenging.

However, we would like to point out that while we were unable to increase WT expression by boosting levels of tRNA^Asn^, we have demonstrated that mutating the N2I directed evolution codon (AUU) to the Ile codon decoded by a rare tRNA^Ile^ (AUA) does diminish protein levels by ~2-fold (Figure 4 —figure supplement 2). This result suggests that tRNA abundance of the second codon may be a contributor to improved Cfr expression, and we state this in the text as follows:

“The isoleucine AUA codon is decoded by the low-abundant tRNA^Ile2^ (Del Tito et al. 1995; Nakamura, Gojobori, and Ikemura 2000). Mutation of N2I from AUU to the AUA rare codon resulted in a ~2-fold decrease in Cfr expression, supporting tRNA abundance as a contributing factor (Figure 4 —figure supplement 2).”

2. The authors find two shortened Cfr isoforms that apparently arise from incorrect initiation on internal methionine codons on cfr mRNA (I26M, M95) and show that translation can start at an initiation codon upstream of M1. This is by itself an exciting story in the Cfr story. The regulatory mechanism due to an uORF is plausible and well documented in Figure 4- supplement 3 and 4. However, the appearance of a (native) internal initiation site starting from M95 is surprising. Given that this is a very prominent translation product (well over 50% with the native Pcfr, Figure 4 – supplement 4), this aspect of the paper raises questions. How does the initiation from M95 depend on antibiotics? The authors clearly show that the shorter isoform does not confer resistance and is rapidly degraded. So, what is the functional role of the shorter isoform? To address these questions the authors could control how much Cfr-short isoform (M95) is present in the absence of antibiotics and how it is affected upon antibiotic treatment. Figures 4-supplementary 3 and 4 are so interesting that I would suggest including them in the main text. The expression of Cfr short (M95) seems to depend on the N2 mutation. This should be quantified and explained.

To address how expression of the truncated product may be impacted by antibiotics, we expressed CfrWT using the Ptet or Pcfr promoter in the presence of sub-inhibitory concentrations of chloramphenicol. Similar results were also obtained with the florfenicol.

Interestingly, we find that levels of the truncated Cfr protein are similar or increased upon incubation with chloramphenicol for both promoter constructs (Author response image 4). While this result is intriguing, it is likely that either increased plasmid copy number or increased Cfr mRNA levels are responsible for this result, as a similar trend is also observed for full-length Cfr. It has been well documented that inhibition of protein synthesis can increase plasmid yields (Begbie et al., 2005; Sambrrok et al., 1989). Furthermore, increased production of the truncation in this context does not impact resistance, as incubation with chloramphenicol increases resistance slightly, likely due to the marginally improved production of full-length Cfr (Author response image 4) . Thus, the functional role of the shorter Cfr isoform remains elusive.

**Author response image 4. sa2fig4:** Impact of antibiotics on expression of full-length and truncated Cfr protein. *E. coli* BW25113 were transformed with pZA constructs containing either CfrWT expressed using the Ptet (WT) or native Pcfr promoter (SCFS1) and plated on LB agar plates containing selection antibiotic and either 0, 0.5, 1, or 2 µg/mL of chloramphenicol (CHL). Overnight cultures were diluted into fresh media containing the respective concentration of CHL and grown to mid-log phase. (a) Cfr protein levels were assessed by immunoblotting against a C-terminal FLAG tag. Asterisks denote the truncated Cfr product. Em = empty vector control. (b) Relative protein expression of full-length Cfr (teal) and the truncation corresponding to translation initiation at Met95 (yellow) compared to CfrWT expression with no CHL. Signal was normalized to housekeeping protein RNA polymerase β-subunit. (c) Minimum inhibitory concentration of CHL required to inhibit bacterial growth of *E. coli* BW25113 transformants that had been pre-incubated with either 0, 0.5, 1, or 2 µg/mL of CHL.

To address how production of the truncation is impacted by Cfr mutations, we have also quantified expression of the truncation in both Cfr variants and in N2 mutants. For Cfr variants containing N2 mutations (CfrV1-5), we observe higher levels of the M95 truncation compared to CfrWT (~2-15 fold, new Figure 2 —figure supplement 4). However, higher expression of the truncation is likely explained by the increased transcript levels for these variants (~1.5-3 fold, Figure 2c). It is well documented that improved translation often results in improved mRNA stability due to increased ribosome occupancy and shielding of the transcript from degradation machinery (Braun et al., 1998; Deana and Belasco, 2005; Edri and Tuller, 2014). In our work, for example, the ~3 fold increase in the CfrV4 transcript level (Figure 2c) is likely explained by its dramatically improved translation observed in polysome analysis (Figure 4d, Figure 4 —figure supplement 1c).

For the N2 mutations alone, we observe higher levels of the M95 truncation for N2K mutants (encoded by AAA or AAG, ~2.5 and ~5 fold respectively, new Figure 3 —figure supplement 1, new Figure 4 —figure supplement 2). Analogous to the Cfr variants, higher expression of the truncation is likely explained by the increased transcript levels of N2K caused by improved translation (Figure 4c). Given that N2I (encoded by AUU or AUA) is not as robustly expressed, and likely not as well translated, the transcriptional impact would be less pronounced, resulting in truncation levels similar to CfrWT (new Figure 3 —figure supplement 1, new Figure 4 —figure supplement 2).

We have edited the text accordingly to incorporate these observations in the Results section:

“The truncations result from translation initiation at internal methionines but do not contribute to resistance (Figure 2 —figure supplement 3), indicating that they are non-functional enzymes unable to methylate A2503. The smaller molecular weight truncation is present in higher levels for all Cfr variants compared to CfrfWT (Figure 2 —figure supplement 4). […] Similarly, to the evolved variants, in addition to the full-length Cfr protein, we also observe expression of the truncation that results from initiation at M95 (Figure 3 —figure supplement 1)”

And in the Discussion section:

“Investigations into expression levels of CfrWT and its respective mutants revealed that, in addition to full-length protein, a smaller Cfr isoform of ~30 kDa is also produced (Figure 2b, Figure 3a). […] Thus, while the truncated product does not contribute to resistance, the potential function of the smaller protein remains elusive and requires further study.”

3. There is a rich literature (cited on p. 21 of the discussion) that may suggest alternative explanations. The authors should identify which of these suggestions can be excluded based on their data and clearly describe those that could contribute to the observed expression effect.The text on p. 21, lines 6-19 should be modified, as it does not exactly state what was actually suggested, there are too many citations without clearly stating what these papers say.

We have edited the discussion to more thoroughly address the potential mechanisms by which N2K may influence translation. The text now reads as follows:

“Of the mutations investigated, N2K is the largest contributor to enhanced Cfr expression and resistance. Although N2K contributes to cellular stability, our results suggest that improved Cfr translation is the dominant role of this mutation. […] Interestingly, the observed internal translation start sites (I26M, M95) that are responsible for producing Cfr truncations (Figure 2b, Figure 2 —figure supplement 3) contain a lysine immediately after methionine, further highlighting the putative role for lysine codons in early steps of translation.”

4. The increased stability of the mutated Cfr variants is fascinating, but not well addressed. What causes the increase in protein stability? What is actually the N-terminus of the protein: is it deformylated and the N-terminal Met cleaved as efficiently as the wt? Is there any correlation with the N-end rule? One way to answer these questions would be to check the N-terminus of proteins from the gels by mass spectrometry. Add a few sentences in the Discussion as to how a (presumably N-terminal) Lys or Ile can increase protein stability.

To address reviewer’s comments, we performed IP-MS/MS analysis of CfrWT and evolved variant CfrV4 which contains the N2K mutation. In brief, we expressed C-terminally FLAG-tagged CfrWT and CfrV4 in *E. coli* and harvested cells at mid-log phase. Cfr protein was immunoprecipitated using anti-FLAG M2 affinity gel (Sigma) from cell lysate under anaerobic conditions. The protein band corresponding to full-length Cfr was excised from the SDS-PAGE gel and subjected to trypsin digestion and subsequent MS/MS analysis. Sequence coverage for CfrWT and CfrV4 are displayed in Author response images 5 and 6.

For CfrWT, we did not detect N-terminal (Nt) peptides with fMet or peptides lacking Met1, suggesting that the Nt Met is retained but deformylated. This result is in line with previous biochemical work demonstrating that residues at the P1’ position larger than valine (V), such as asparagine (N), are disfavored substrates for methionine aminopeptidase (Frottin et al., 2006; Hirel et al., 1989; Xiao et al., 2010)

Due to high lysine content at the N-terminus of CfrV4, we were unfortunately unable to detect Nt peptides for the variant protein. However, previous biochemical work has demonstrated that basic side-chains such as lysine (K) at position 2 are slightly favored substrates for peptide deformylase (Ragusa et al., 1999) and strongly disfavored for methionine aminopeptidase (Frottin et al., 2006; Hirel et al., 1989). Similarly, nascent peptides with isoleucine (I) at position 2 are also likely to be efficiency deformylated and resistance to Met excision (Frottin et al., 2006; Hirel et al., 1989; Xiao et al., 2010).

**Author response image 5. sa2fig5:** MS/MS peptide coverage for CfrWT.

**Author response image 6. sa2fig6:** MS/MS peptide coverage for CfrV4.

Although we cannot completely rule out its involvement in stability, these preliminary results and existing biochemical evidence suggest that differential co-translational processing of the N-terminus, which would invoke ”classic” N-end rule or fMet/N-degron pathways (Izert et al., 2021), is likely not responsible for the observed improved stability of N2K/I mutants.

Although the precise mechanism by which N2K/I improve Cfr stability remains elusive, these mutants could alter recognition by other enzymes that may be important in regulating degradation, such as endopeptidases or L/F-tRNA-protein transferase (LFTR). For example, it was discovered that LFTR can add primary destabilizing residues to Nt Met, creating an N-degron (Dougan et al., 2012), but its efficiency can depend on the identity of the second amino acid (Ottofuelling et al., 2021).

We have modified the text to include discussion on how N2K/I may be involved in protein stability, which now reads as follows.

“We also observe modest improvements in protein stability with N2K/I mutants (Figure 4e). In bacteria, the identity of N-terminal residues are important determinants of degradation through N-degron pathways (Dougan, Micevski, and Truscott 2012; Tobias et al. 1991). During protein synthesis, the N-terminus is co-translationally processed by two enzymes, peptide deformylase to remove the formyl group from Met (fM) and methionine aminopeptidase (Koubek et al. 2021). Based on previous biochemical work, it is unlikely that CfrWT and CfrN2K/I would have different N-terminal processing, since fMN… and fMK/I… are likely to be efficiently de-formylated (Ragusa et al. 1999) and resistant to methionine excision (Hirel et al. 1989; Frottin et al. 2006; Xiao et al. 2010). Although the precise mechanism by which N2K/I improves Cfr stability remains elusive, these mutations may alter recognition by other enzymes important for degradation, such as endopeptidases or L/F-tRNA-protein transferase (Izert, Klimecka, and Górna 2021; Ottofuelling et al. 2021).”

5. Cfr methylates C8 of A2503. Mass spectra clearly distinguish the peaks of m2m8A2503 and m2A2503. However, this appears to rely on the assumption that A2503 is completely m2-modified. Has this been checked? What would be the mass of the unmodified fragment and was it identified in the spectra? Can the authors exclude the appearance of m8A2503 species? Please comment on the modification completeness of m2A2503 and provide the expected m/z value for the respective unmodified RNA fragment.

The RNA fragment containing unmodified A2503 (AΨG) has a predicted m/z value of 999.14 (now included in Figure 2 —figure supplement 1). The mass of the unmodified fragment overlaps with an unrelated ACG fragment which has a predicted m/z value of 998.16 (see detailed m/z predictions obtained from ChemDraw, Author response image 7). Since the mass of the unmodified A2503 is near-identical to the mass + 1 isotope of the unrelated fragment (999.14 and 999.16, respectively), it cannot be accurately determined if unmodified A2503 is present our MALDI-MS samples. However, the predicted isotope abundance for the unrelated fragment (37%) is qualitatively similar in spectra obtained from wild type *E. coli* with endogenous RlmN only (Author response image 8) and suggests little-to-no presence of the unmodified AΨG species. Although we cannot conclude definitively, these results suggest that RlmN achieves complete or near-complete C2 methylation of A2503. Relative abundance of the isotope peak is also consistent in samples where CfrWT or Cfr variants were expressed, suggesting that there is likely little/no unmodified A2503 present in these samples either.

**Author response image 7. sa2fig7:** m/z predictions obtained from ChemDraw for rRNA fragments derived from oligo-protection MALDI-TOF mass spectrometry. The predicted m/z and abundance for the mass + 1 isotope peak is labeled in green.

**Author response image 8. sa2fig8:** Annotated MALDI spectra for empty vector control, CfrWT/V4/V7. The mass peak (m) is labeled in blue, while the isotope peak (m+1) and predicted relative abundance derived from ChemDraw is labeled in green.

While the monomethylated mass of 1013 could also correspond to m^8^A2503, this scenario is highly unlikely. This is because m^8^ modification would induce strong antibiotic resistance, as shown by our data, which is not what we observed in samples where 1013 predominates. Additionally, our previous work indicates that RlmN methylates A2503 in early stages of ribosome biogenesis.

6. Since the ribosome structure presented is of a modified *E. coli* ribosome and the cfr enzyme is originally from *S. aureus*, the sequence alignment of Cfr from *S. aureus* and *E. coli* should be presented (as a Supplement). The authors should discuss whether it is legitimate to use *S. aureus* cfr to modify *E. coli* ribosome. This would widen the scope of the paper by generalizing the conclusions for gram-positive and gram-negative bacteria. This could be addressed in the introduction (e.g., p. 4 line 9 and line 15) and in the Results section (e.g., p.16 line 4 and line 9). In Figure 5 – add to the legend that this is an *E. coli* ribosome. In addition, please relate to this point in the discussion (p. 20).

The *cfr* gene recovered from animal-derived *E. coli* isolates (Deng et al., 2014; Liu et al., 2017; Ma et al., 2021; Wang et al., 2012) is identical in sequence to the clinical *S. aureus* isolate used in this study. Conservation of the sequence, along with the observation that this Cfr sequence confers resistance in veterinary *E. coli* isolates, validates the use of *E. coli* as a model organism for our studies.

We have modified the text to clarify this point in the introduction accordingly:

“Since then, the *cfr* gene has been identified across the globe in both gram-positive and gram-negative bacteria, including *E. coli* (Shen, Wang, and Schwarz 2013; Vester 2018)…”

Although Cfr is predominantly found in *Staphylococcal* species, the 23S rRNA sequence is highly conserved between *E. coli* and *S. aureus* (Eyal et al., 2015). Specifically, the sequence and structure of the PTC is highly conserved, which we have now illustrated by including a structural overlay of the *E. coli* Cfr-modified ribosome and *wild-type S. aureus* ribosome as Figure 5 —figure supplement 3.

We have also added the following text within the Results section as follows:

“Although Cfr has been identified in animal-derived *E. coli* isolates (Wang et al. 2012; Deng et al. 2014; Liu et al. 2017; Ma et al. 2021), the resistance gene has primarily been identified clinically in staphylococcal organisms such as *S. aureus* (Vester 2018). However, given the high sequence and structural conservation within the PTC region (Figure 5 —figure supplement 3), structural impacts of the Cfr m^8^A2503 modification within *E. coli* and *S. aureus* ribosomes are likely conserved.”

7. The authors superimposed the complex structure of an archeal ribosome and linezolid and tiamulin for explaining the resistance mechanism. However, this resistance mechanism was not found in archaea and thus these ribosomes are less relevant for clinical aspects. More appropriate models to be used are of bacterial ribosomes in complex with these antibiotics, i.e. D. radiodurans ribosome in complex with tiamulin (PDBID 1XBP) and the pathogen *S. aureus* ribosome in complex with linezolid (PDBID 4WFA).

Previous SAR work has demonstrated that the C5 group of linezolid is likely responsible for steric collision with m^8^A2503. However, in the three published structures, the C5 group is modeled in dramatically different conformations (outlined in Author response image 9). Although the structure of the *S. aureus* ribosome in complex with linezolid (PDB: 4WFA, purple) represents a more biologically-relevant organism, the modeled conformation of the C5 group, for which there is weak density, does not sterically clash with m^8^A2503. For the remaining structures (PDB: 3CPW in yellow and PDB: 3DLL in pink), while both modeled C5 conformations would sterically clash with m^8^A2503, 3CPW exhibits superior density for the C5 moiety. We also recently obtained a structure of a linezolid-bound *E. coli* ribosome (Tsai et al., 2021), and the C5 group is best modeled in a conformation near-identical to 3CPW. Together, these results, combined with the observation that Cfr confers antibiotic resistance in *E. coli*, support our decision to use the *H. marismortui* ribosome (PDB: 3CPW) for structural comparison.

**Author response image 9. sa2fig9:** 

A2503 is modeled in the *syn-*conformation in the *H. marismortui* ribosome-tiamulin complex (PDB: 3G4S, purple)*,* the same conformation as m^8^A2503 in the Cfr-modified ribosome. A2503 is modeled as *anti-* in the *D. radiodurans* structure (PDB: 1XBP, yellow). Thus, the archaeal structure was chosen to provide more straight-forward structural comparison. Superposition of both tiamulin-ribosome complexes reveals that m^8^A2503 sterically clashes with the C10 and C11 substituents of tiamulin in both cases.

**Author response image 10. sa2fig10:** 

8. The paper would benefit from an analysis of the natural variability of the evolved sites. It is mentioned that clinical isolates do not have these exact substitutions, but a supplemental multiple sequence alignment of homologues with these positions marked would anyway be interesting. For example, an N2K variant appears in the results of a quick blastp search in a Bacillus Cfr.

We analyzed sequences of Cfr homologues with less than 80% sequence identity and did discover that directed evolution mutations are recapitulated in these sequences. We have added a supplementary figure and a section to the discussion which reads as follows:

“Interestingly, mutations obtained through directed evolution have been observed in Cfr homologues that share less than 80% sequence identity with Cfr. Specifically, methionine (M) at position 26 is observed for the functionally characterized Cfr homologues Cfr(B) (Deshpande et al. 2015; Marín et al. 2015; Hansen and Vester 2015) and Cfr(D) (Pang et al. 2020), which have been recovered from human-derived isolates and share 74% and 64% amino acid identity with Cfr, respectively (Schwarz et al. 2021) (Figure 4 —figure supplement 5). We also observe lysine (K) at position 2, methionine (M) at position 26, and glycine (G) at position 39, akin to N2K, I26M, and S39G mutations, for a number of Cfr homologues that clade with functional Cfr or Cfr-like genes (Stojković et al. 2019). While the precise roles of these residues within less-well characterized and more distantly related Cfr proteins requires further study, these observations indicate that directed evolution accessed sequence space that is already being exploited by proteins that are, or are hypothesized to be, functional Cfr resistance enzymes.”

9. Another remaining question is how the novel variants affects susceptibilities to ribosome-targeting antibiotics in other structural classes, including nucleoside analogues and macrolides. In the introduction the authors mention that in addition to PhLOPSA antibiotics, Cfr can confer resistance to nucleoside analog A201A, hygromycin A, and 16-membered macrolides. However, this is not returned to in the paper. MIC experiments with an expanded selection of antibiotics would be really nice to see if possible. Similarly, does the structure explain the previously observed cross-resistance that goes beyond PhLOPS(A)?

We agree with the reviewer and performed MIC testing of hygromycin A against *E. coli* BW25113 lacking efflux component *acrB* (Figure 1 —figure supplement 1). Analogous to PhLOPS_A_ antibiotics, Cfr variants confer higher resistance to hygromycin A than CfrWT (4-fold higher resistance for CfrV3 and 8-fold higher resistance for CfrV7).

We also performed additional structural overlays of the Cfr-modified ribosome with existing structures of ribosomes in complex with hygromycin A, nucleoside analog A201A, and 16-membered macrolides tylosin and josamycin (Figure 5 —figure supplement 4). As expected, the Cfr modification introduces a steric clash between the C8 methylation mark and the antibiotics, providing molecular rationale for how Cfr confers resistance to, in total, 8 classes of ribosome-targeting antibiotics.

References

Begbie S, Dominelli G, Duthie K, Holland J, Jitratkosol M. 2005. The effects of sub-inhibitory levels of chloramphenicol on pBR322 plasmid copy number in *Escherichia coli* DH5α cells. *J Exp Microbiol Immunol* 7:82–88.

Braun F, Le Derout J, Régnier P. 1998. Ribosomes inhibit an RNase E cleavage which induces the decay of the rpsO mRNA of *Escherichia coli*. *EMBO J* 17:4790–4797.

Deana A, Belasco JG. 2005. Lost in translation: the influence of ribosomes on bacterial mRNA decay. *Genes Dev* 19:2526–2533.

Dégut C, Monod A, Brachet F, Crépin T, Tisné C. 2016. in vitro/in vivo Production of tRNA for X-Ray Studies. *Methods Mol Biol* 1320:37–57.

Deng H, Sun J, Ma J, Li L, Fang L-X, Zhang Q, Liu Y-H, Liao X-P. 2014. Identification of the multi-resistance gene cfr in *Escherichia coli* isolates of animal origin. *PLoS One* 9:e102378.

Dougan DA, Micevski D, Truscott KN. 2012. The N-end rule pathway: From recognition by N-recognins, to destruction by AAA proteases. Biochimica et Biophysica Acta (BBA) – Molecular Cell Research. doi:10.1016/j.bbamcr.2011.07.002

Edri S, Tuller T. 2014. Quantifying the effect of ribosomal density on mRNA stability. *PLoS One* 9:e102308.

Eyal Z, Matzov D, Krupkin M, Wekselman I, Paukner S, Zimmerman E, Rozenberg H, Bashan A, Yonath A. 2015. Structural insights into species-specific features of the ribosome from the pathogen *Staphylococcus aureus*. *Proc Natl Acad Sci U S A* 112:E5805–14.

Frottin F, Martinez A, Peynot P, Mitra S, Holz RC, Giglione C, Meinnel T. 2006. The Proteomics of N-terminal Methionine Cleavage* S. *Mol Cell Proteomics* 5:2336–2349.

Hirel PH, Schmitter MJ, Dessen P, Fayat G, Blanquet S. 1989. Extent of N-terminal methionine excision from *Escherichia coli* proteins is governed by the side-chain length of the penultimate amino acid. *Proc Natl Acad Sci U S A* 86:8247–8251.

Izert MA, Klimecka MM, Górna MW. 2021. Applications of Bacterial Degrons and Degraders — Toward Targeted Protein Degradation in Bacteria. *Frontiers in Molecular Biosciences*. doi:10.3389/fmolb.2021.669762

Kramer EB, Farabaugh PJ. 2007. The frequency of translational misreading errors in *E. coli* is largely determined by tRNA competition. *RNA* 13:87–96.

Liu X-Q, Wang J, Li W, Zhao L-Q, Lu Y, Liu J-H, Zeng Z-L. 2017. Distribution of cfr in Staphylococcus spp. and *Escherichia coli* Strains from Pig Farms in China and Characterization of a Novel cfr-Carrying F43:A-:B- Plasmid. *Frontiers in Microbiology*. doi:10.3389/fmicb.2017.00329

Ma Z, Liu J, Chen L, Liu X, Xiong W, Liu J-H, Zeng Z. 2021. Rapid Increase in the IS26-Mediated cfr Gene in *E. coli* Isolates with IncP and IncX4 Plasmids and Co-Existing cfr and mcr-1 Genes in a Swine Farm. *Pathogens* 10. doi:10.3390/pathogens10010033

Ottofuelling RD, Ninnis RL, Truscott KN, Dougan DA. 2021. Novel modification by L/F-tRNA-protein transferase (LFTR) generates a Leu/N-degron ligand in *Escherichia coli*. *bioRxiv*.

Ragusa S, Mouchet P, Lazennec C, Dive V, Meinnel T. 1999. Substrate recognition and selectivity of peptide deformylase. Similarities and differences with metzincins and thermolysin. *J Mol Biol* 289:1445–1457.

Sambrrok J, Fritsch EF, Maniatis T. 1989. Molecular cloning a laboratory manual 2nd edition.

Shepherd J, Ibba M. 2015. Bacterial transfer RNAs. *FEMS Microbiol Rev* 39:280–300.

Tsai K, Stojković V, John Lee D, Young ID, Szal T, Vazquez-Laslop N, Mankin AS, Fraser JS, Fujimori DG. 2021. Structural basis for context-specific inhibition of translation by oxazolidinone antibiotics. *bioRxiv*. doi:10.1101/2021.08.10.455846

Wang Y, He T, Schwarz S, Zhou D, Shen Z, Wu C, Wang Y, Ma L, Zhang Q, Shen J. 2012. Detection of the staphylococcal multiresistance gene cfr in *Escherichia coli* of domestic-animal origin. *J Antimicrob Chemother* 67:1094–1098.

Xiao Q, Zhang F, Nacev BA, Liu JO, Pei D. 2010. Protein N-terminal processing: substrate specificity of *Escherichia coli* and human methionine aminopeptidases. *Biochemistry* 49:5588–5599.